# Dopaminergic processes predict temporal distortions in event memory

**Erin Morrow, Ringo Huang ⓘ & David Clewett ⓘ ✉**

Our memories do not simply keep time − they distort it, stretching and compressing the past to reflect the structure of experience. Here, we combined functional magnetic resonance imaging (fMRI; $n = 32$) with eye-tracking ($n = 28$) to test whether activation of the dopaminergic system, known to influence encoding and time perception, expands mnemonic representations of time between contextually distinct events. Participants encoded item sequences while listening to tones that typically repeated over time, but occasionally changed, creating salient event boundaries. We found that tone switches significantly activated the ventral tegmental area (VTA), and the magnitude of these responses predicted greater time dilation between item pairs spanning those switches. At a longer timescale, increased blinking also predicted greater time dilation in memory, but only for boundary-spanning item pairs. Together, these findings suggest that dopaminergic processes are sensitive to event structure and contribute to distortions of remembered time that may help segment continuous experience into distinct episodic memories.

"[Mechanical time] is unyielding, predetermined. [Body time] makes up its mind as it goes along," remarks physicist and author Alan Lightman (p. 32)[1]. Indeed, time can seem malleable when we reflect upon the past. For example, experiences from the COVID-19 pandemic are not equally represented in memory. The early lockdown period is often recalled as having passed more quickly than it was originally experienced, potentially driven by a significant lack of variety or novelty[2,3]. In contrast, everyday life typically involves frequent shifts in context, or event boundaries, which tend to stretch remembered time and promote the separation of experiences into distinct memories. These memory-expanding effects have been reliably demonstrated in laboratory settings, with changes in scenes[4], goals[5], and emotional states[6,7] eliciting time dilation in long-term memory. Yet, despite their central role in human cognition, it remains unclear how the brain uses contextual cues to reshape temporal representations in service of memory organization.

A promising candidate for this process is the ventral tegmental area (VTA), a brainstem nucleus that releases dopamine and modulates episodic memory. Dopamine facilitates encoding and consolidation of behaviorally relevant information over both short[8] and long timescales[9,10]. The VTA responds to a wide variety of salient stimuli, including those that are novel[11] and carry predictive salience[12]. Neuroimaging studies in humans suggest that phasic VTA activation and dopamine may signal key moments when ongoing predictions fail and it is beneficial to update one's mental model of what is happening[13,14]. Indeed, this framework is consistent with dopamine's known role in working memory, which includes modulation of striatal activity to update ongoing representations[15] held in short-term memory storage[16]. Additionally, midbrain dopamine regulates synaptic plasticity in the hippocampus[12,17], a region that triggers the encoding of new memories and their temporal features[18]. Together, these factors make the VTA well equipped to mediate the influence of event boundaries on the temporal structure of memory.

Alongside its role in memory, dopamine is heavily implicated in time perception. According to the dopamine clock hypothesis[19,20], transient increases in dopamine may accelerate a hypothetical internal clock that generates pulses, or ticks, that are collected and stored by a memory system to estimate time. The more pulses collected during a fixed interval, the longer that period is perceived to last[21]. Thus, dopamine-induced acceleration of this clock would result in time

Department of Psychology, University of California, Los Angeles, CA, USA. ✉e-mail: david.clewett@psych.ucla.edu

overestimation[22]. Other frameworks offer additional nuance; for example, dopamine may modulate this clock based on prediction errors[22,23], which are theorized to trigger event segmentation[24]. Many timing models also contain a working memory storage component that affects time perception[22] and is influenced by dopaminergic processes. However, most research has focused on how dopamine influences time perception as an experience unfolds, using tasks in which participants actively track the duration of an ongoing stimulus or estimate duration immediately after the stimulus ends. Therefore, it remains unclear whether dopamine also helps to encode exaggerated representations of time into long-term memory.

Intriguingly, blink behavior may offer an indirect window into the influence of dopamine and event boundaries on temporal memory distortions. Recent neuroimaging work shows that momentary blinks correspond with increased activation in dopaminergic regions, including the VTA and substantia nigra[25]. Moreover, it has been shown that spontaneous blink rate is altered in individuals with dopamine-related neuropsychiatric conditions like Parkinson's disease[26] and schizophrenia[27] under certain conditions[28,29]. Like dopamine, blinks also tend to occur around breakpoints in experience[30,31], such as punctuation marks in text passages[32], pauses in speech[33], and the beginnings of quiz questions on a mentally demanding TV game show[34]. Additionally, one study in humans showed that increased blinking during a short interval was associated with exaggerated estimates of time[35], consistent with the role of dopamine in shaping time perception.

In the current neuroimaging and eye-tracking study, we find that context shifts during item sequences predict increases in two indirect measures of dopaminergic processing: VTA activation and blinking. Irrespective of contextual stability or change, activation of the VTA is tightly coupled with momentary increases in blinking, supporting the putative relationship between dopamine and blink behavior. We also find that VTA activation and more prolonged periods of blinking across event boundaries predict the later expansion of temporal representations in memory. These findings extend the relationship between dopamine and time perception to episodic memory, revealing how the brain not only remembers time, but may also bend it to represent contextually distinct events.

## Results

### Event boundaries predicted subjective time dilation effects in long-term memory, a behavioral index of event separation

Participants performed a modified version of a paradigm known to induce temporal distortions in event memory[36]. Participants studied sequences of neutral object images, each preceded by a 1 s pure tone presented to their left or right ear (Fig. 1). The laterality of the tone remained the same for eight items, creating the perception of a stable context or event. However, after 8 successive items in each sequence, the tone abruptly switched to the other ear and changed pitch, forming a perceptual event boundary. Participants were also instructed to switch the hand they used to respond to an orienting question for each image (Is this object larger or smaller than a standard shoebox?; for details on response distributions, see Supplementary Fig. 3). In this way, event boundaries were not only perceptually salient but also task relevant. Each sequence contained three event boundaries, leading to four 8-item events.

Following each sequence, participants completed two temporal memory tests for different pairs of objects: a temporal order test and a temporal distance rating test. Both these tests were designed to assess event separation effects in long-term memory. Because we were specifically interested in changes in subjective temporal memory, we focused on analyzing data from the distance ratings task. The results of

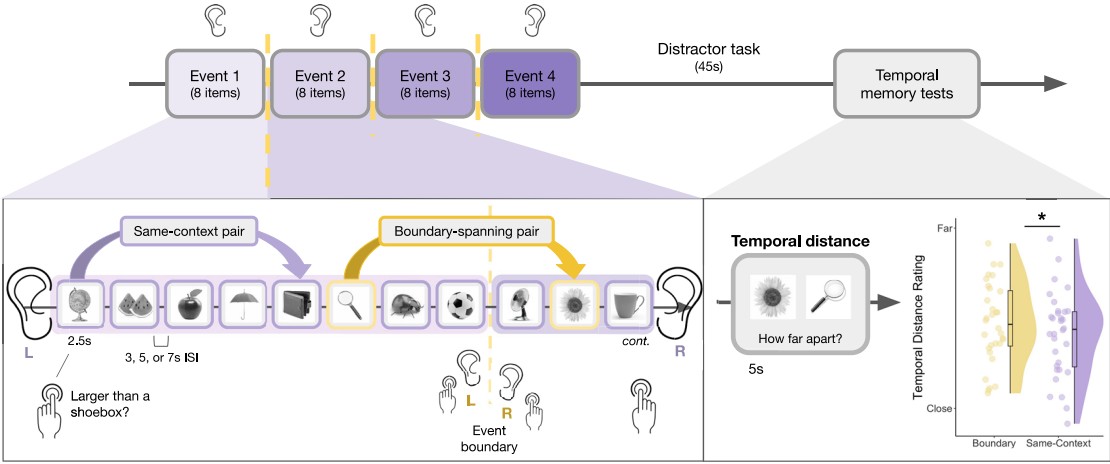

**Fig. 1 | Event sequence encoding task and temporal distance memory test.** In each block of the task, participants studied a sequence of 32 neutral object images. Prior to each image, participants listened to pure 1 s tones either in their left or right ear during the midpoint of a fixation screen. The side of the tone also cued participants as to which hand they should use to judge whether each object was larger than a shoebox (e.g., left ear = left hand). The image sequence was divided into four events. Events were defined by a stable auditory context, with the same tone presented to the same ear for 8 successive items. After 8 items, the tone switched to the other ear, representing an event boundary (see dashed yellow line). Additionally, the pitch of the tone changed, as well as the hand required by the participants to make button responses. This pattern continued for the remainder of the 8-item event. After four events and a brief distractor task, participants completed a temporal distance memory test (along with a temporal order memory test discussed elsewhere[37]). On each test trial, participants were presented with different pairs of objects from the prior sequence and asked to rate how far apart they thought each pair appeared in time (very close, close, far, or very far), despite their equivalent objective distance. Some tested pairs had been encountered in the same context (in purple), while other pairs had been encountered with an intervening boundary (in yellow). In right panel, raincloud plot displays distance memory ratings for boundary-spanning and same-context pairs. Each dot represents averaged data from one participant, with distance memory ratings (1–4) displayed as a continuous variable. Colored boxplots represent 25th–75th percentiles of the data, and the center line represents the median. Statistical significance refers to results from trial-level cumulative link model. ISI = interstimulus-interval. *$p < 0.05$. Source data are provided as a Source Data file. The images in this figure are credited to: Alex Staroseltsev/Shutterstock.com; GalapagosPhoto/Shutterstock.com; Bildagentur Zoonar GmbH/Shutterstock.com; Smileus/Shutterstock.com; Sergiy1975/Shutterstock.com; Anton Starikov/Shutterstock.com; James.Pintar/Shutterstock.com; Preto Perola/Shutterstock.com; tanadtha lomakul/Shutterstock.com; Aleksey Sagitov/Shutterstock.com; and iStock.com/ramzihachicho.

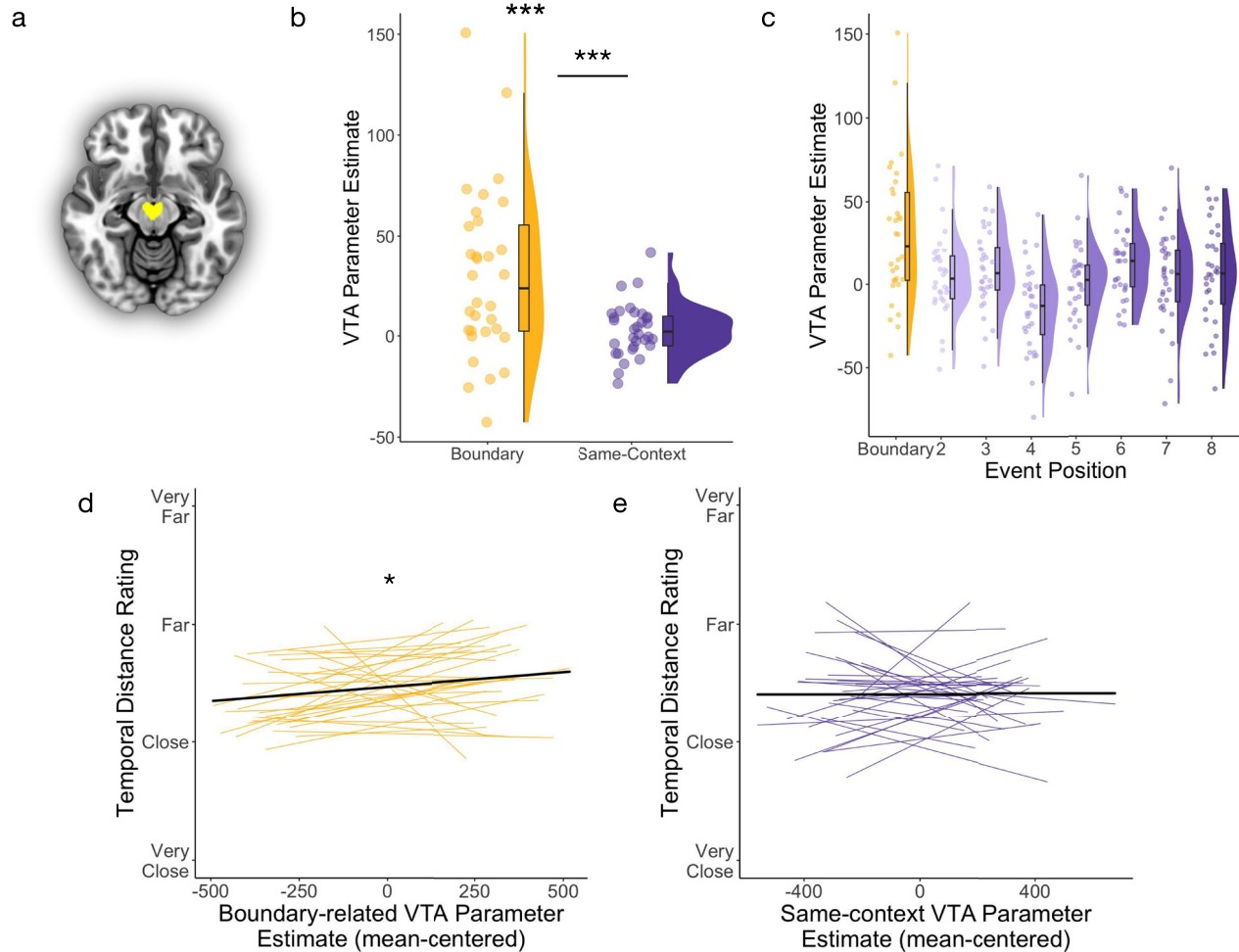

**Fig. 2 | Event boundaries predicted greater VTA activation, and these responses predicted greater time dilation effects in memory. a** Axial slice of the probabilistic atlas used to identify the VTA (yellow), thresholded at 75% tissue-type probability[66]. **b** VTA parameter estimates, a measure of brain activation levels, for boundary tones (yellow) and same-context tones (purple). Each dot represents averaged data from one participant (*n* = 32 participants). For box plots, boxes are defined by 25th percentile, median, and 75th percentile. Whiskers are defined by minimum and maximum, excluding outliers. Statistical significance refers to results from trial-level linear mixed effects models (boundary > baseline: *p* = 0.00034; boundary > same-context: *p* = 0.00000098). **c** VTA parameter estimates plotted by trial position within the 8-item events during encoding. The first tone in each 8-item event was a boundary trial, when the tone switched from one ear to the other.

Same-context tones then played in the same ear and were repeated across item positions 2 through 8 within that stable auditory event. Each dot represents averaged data from one participant (*n* = 32 participants). For box plots, boxes are defined by 25th percentile, median, and 75th percentile. Whiskers are defined by minimum and maximum, excluding outliers. **d** Participant-level trendlines plotting the relationship between boundary-related VTA parameter estimates (mean-centered within condition) and temporal distance memory ratings (displayed as a continuous variable). Statistical significance refers to results from trial-level cumulative link model (*p* = 0.034). **e** Participant-level trendlines plotting the relationship between same-context VTA parameter estimates and temporal distance memory ratings. Dark, bold lines represent the average linear trend across participants. ****p* < 0.001; **p* < 0.05. Source data are provided as a Source Data file.

the temporal order test are reported elsewhere[37]. In the temporal distance ratings task, participants were asked how far apart they remembered two images occurring in the previous sequence, ranging from very close to very far apart. Importantly, there were always three intervening images (approximately 32.5 s) between each tested pair, making their objective distance identical. Thus, any differences in distance memory ratings were due to subjective distortions in remembered time.

Replicating prior findings, boundary-spanning pairs were remembered as having been encountered farther apart in time than same-context pairs (ß = 0.071, *z* = 2.30, *p* = 0.022; odds ratio = 1.07, 95% CI of odds ratio = [1.01, 1.14]); Fig. 1, bottom right panel), consistent with the idea that boundaries serve to mentally distance contextually distinct episodes in memory. For more detail on participants' distance response distributions, see Supplementary Fig. 4. Interestingly, we also found a significant main effect of pair position (*z* = 4.31, *p* < 0.001; odds

ratio = 1.04, 95% CI of odds ratio = [1.02, 1.05]), such that later pair positions in the lists were remembered as occurring farther apart in time than item pairs encountered early in the list (see Supplementary Fig. 5). This pattern suggests a time-on-task-like effect for subjective distance ratings as the sequence unfolded.

### Event boundaries reliably predicted BOLD activation in the VTA, and these brain responses predicted later time dilation effects in memory

In the next analysis, we tested our key hypothesis that event boundaries, or tone switches, engage VTA activation during encoding (see Fig. 2a for anatomical mask). Consistent with our prediction, we found that Tone Type significantly predicted VTA activation, such that boundary tones elicited higher VTA activation compared to same-context tones (*t*(9281.88) = 4.90, *p* < 0.001; ß = 0.09, 95% CI = [0.05, 12]). When examining each Tone Type separately, we found that VTA

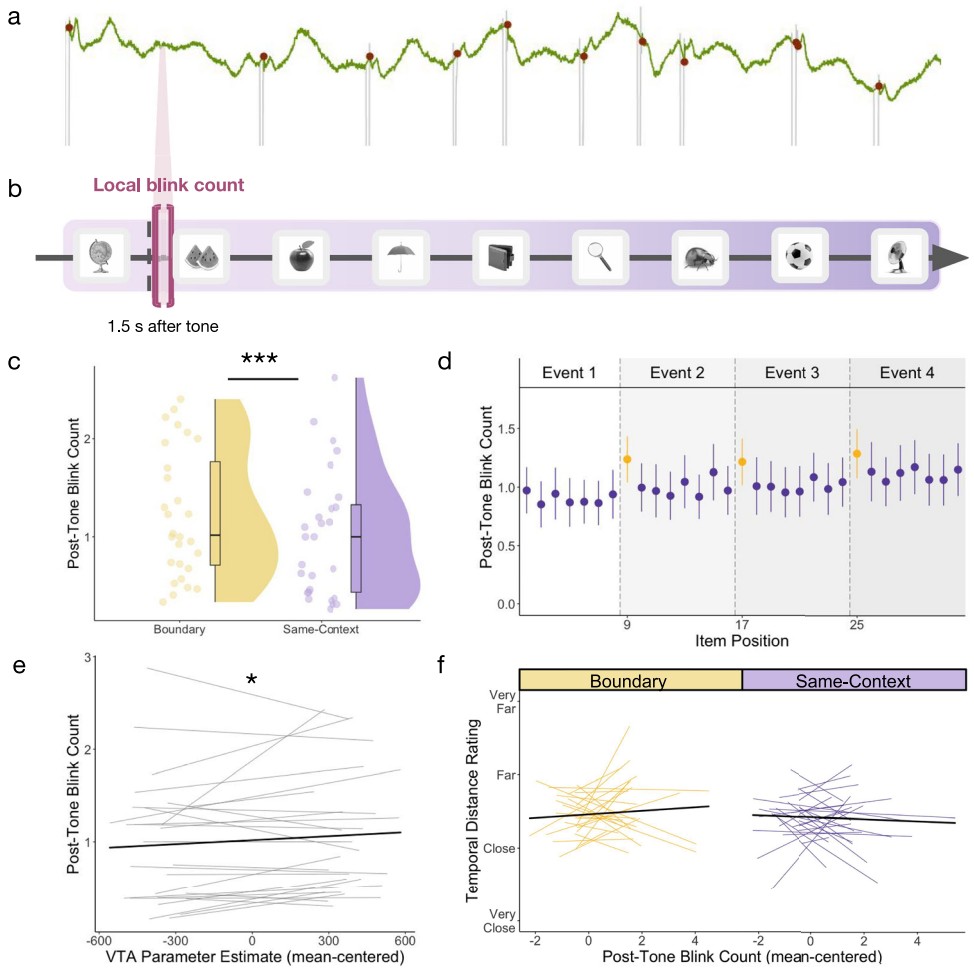

**Fig. 3 | Event boundaries predicted a momentary increase in blinking, and these local changes in blinking were predicted by tone-related VTA activation irrespective of contextual stability or change. a** Example pupil timeseries for a subset of an encoding block. Green line displays pupil size (in arbitrary units) over time. Red circles represent time points when blink artifacts were identified by the pupil preprocessing algorithm based on velocity changes in pupil size. Timeseries is displayed for illustrative purposes and does not align with the trial schematic. **b** Schematic of the post-tone blink period of interest during event sequence encoding. Local blink count was captured in the short 1.5 s interval following each tone onset (bracketed in pink). **c** Overall comparison between mean blink count after the onset of boundary (yellow) and same-context tones (purple). Each dot represents averaged data from one participant (*n* = 28 participants). For box plots, boxes are defined by 25th percentile, median, and 75th percentile. Whiskers are defined by minimum and maximum, excluding outliers. Statistical significance refers to results from trial-level linear mixed effects model (*p* = 0.0000000014). **d** Mean post-tone blink count by item position in each encoding block (2-32; first item excluded). Yellow dots represent blink counts after boundary tones, which always occurred every 8 tones (preceding items 9, 17, and 25). Purple dots represent

blink counts after same-context tones, which occurred in all remaining positions in the lists. Error bars represent SEM. *n* = 28 participants. **e** Participant-level trendlines plotting the relationship between trial-level VTA parameter estimates (mean-centered) and post-tone blink count. The dark, bold line represents the average linear trend across participants. Statistical significance refers to results from trial-level linear mixed effects model (*p* = 0.038). **f** Participant-level trendlines plotting the relationship between post-tone blink count (mean-centered) and temporal distance ratings for boundary-spanning pairs (yellow; left) and same-context pairs (purple; right). For modeling, post-tone blink counts are mean centered for each condition separately. The dark, bold lines represent the average trendline across participants, also for each condition separately. ***p < 0.001; *p <0.05. Source data are provided as a Source Data file. The images in this figure are credited to: Alex Staroseltsev/Shutterstock.com; GalapagosPhoto/Shutterstock.com; Bildagentur Zoonar GmbH/Shutterstock.com; Sergiy1975/Shutterstock.com; Anton Starikov/Shutterstock.com; James.Pintar/Shutterstock.com; Preto Perola/Shutterstock.com; tanadtha lomakul/Shutterstock.com; Aleksey Sagitov/Shutterstock.com; and iStock.com/ramzihachicho.

activation was significantly greater than baseline for boundary tones ($t(31.23) = 4.02$, $p < 0.001$; $\beta_0 = 30.04$, 95% CI = [15.37, 44.71]), but not for same-context tones ($t(30.29) = 1.45$, $p = 0.16$; $\beta_0 = 3.35$, 95% CI = [−1.17, 7.88]) (Fig. 2b, c).

We next tested if these boundary-related VTA responses predicted increased temporal distance ratings between item pairs spanning those same boundary tones. The results revealed no significant interaction effect of VTA activation and Pair Type (boundary-spanning versus same-context pair) on temporal distance memory ($z = 1.40$, $p = 0.16$; odds ratio = 1.05, 95% CI of odds ratio = [0.98, 1.11]). However, when analyzing each Pair Type separately, we found that boundary-evoked VTA activation significantly predicted higher temporal

distance ratings ($z = 2.13$, $p = 0.034$; odds ratio = 1.11, 95% CI of odds ratio = [1.01, 1.22]) (Fig. 2d). In contrast, tone-related VTA activation did not predict temporal distance ratings for the same-context pairs spanning those tones ($z = 0.073$, $p = 0.94$; odds ratio = 1.00, 95% CI of odds ratio = [0.93, 1.09]) (Fig. 2e). Thus, event boundaries appear to engage the VTA in a way that may distort the temporal structure of memory.

**Event boundaries predicted a momentary increase in blinking**

Intriguingly, prior studies have shown that breakpoints in experience are associated with transient increases in spontaneous blinking[31,34], which in turn has been linked to dopaminergic processes[28]. We sought

to replicate these results in the current paradigm by examining how tone switches predict transient changes in blinking (Fig. 3a, b). This model revealed a main effect of Tone Type on local blink count ($t$(7461) = 6.07, $p < 0.001$; $\beta = 0.10$, 95% CI = [0.07, 13]), such that boundary tones were immediately followed by more blinks than same-context tones (Fig. 3c, d).

### Tone-related increases in VTA activation predicted blinking irrespective of contextual stability

To validate the putative link between blink behavior and dopaminergic processes, we next examined trial-level coupling between post-tone blink count and tone-related VTA activation. We found a significant main effect of VTA activation on post-tone blink count ($t$(7270) = 2.08, $p = 0.038$; $\beta = 0.03$, 95% CI = [0.0019, 07]), such that greater tone-related VTA activation was associated with higher blink count irrespective of condition (Fig. 3e). There was no significant VTA-by-tone type interaction effect on post-tone blink count ($t$(7271) = 1.34, $p = 0.18$; $\beta = 0.02$, 95% CI = [−0.01, 05]), reinforcing a more general, context-invariant association between phasic VTA activation and blinking.

To verify that blink changes likely resulted from tone-related VTA activation (and not vice versa), we also performed a control analysis focusing on blinks that occurred within the 1.5 s period *before* each tone. Pre-tone blink count was not significantly predicted by Tone Type ($t$(7410) = 0.83, $p = 0.41$; $\beta = 0.01$, 95% CI = [−0.02, 05]) or tone-related VTA activation ($t$(7224) = 0.95, $p = 0.34$; $\beta = 0.02$, 95% CI = [−0.02, 05]). Together, these blink timing-related control analyses provided strong evidence that blinking was triggered by behaviorally relevant changes during a dynamic experience.

### Momentary increases in local boundary-evoked blinking behavior were not associated with later temporal distance ratings in memory

Finally, we tested if post-tone blink count was related to later memory separation effects. Contrary to our prediction, there was no main effect of post-tone blink count ($z = 0.50$, $p = 0.62$; odds ratio = 1.02, 95% CI of odds ratio = [0.95, 1.10]) or blink-by-pair type interaction effect on temporal distance ratings ($z = 1.54$, $p = 0.12$; odds ratio = 1.06, 95% CI of odds ratio = [0.98, 1.15]) (Fig. 3f).

### Testing relations between more prolonged periods of blinking, VTA activation, and temporal distance memory

Beyond local changes in blinking around boundaries, we were also interested in testing whether blinking across larger timescales (i.e., windows spanning approximately 32.5 s between to-be-tested item pairs) predicted distance memory and VTA activation (Fig. 4a, b).

### Increased blinking between to-be-tested item pairs was related to tone-related VTA activation and time dilation effects in memory

First, we tested whether boundaries modulated blink count across these larger time periods. We found that Pair Type (boundary-spanning pair versus same-context pair) did not significantly predict blink count across the to-be-tested pair windows ($t$(2881.02) = 1.55, $p = 0.12$; $\beta = 0.02$, 95% CI = [−0.0050, 04]) (Fig. 4c, d).

However, tone-related VTA activation significantly predicted blink count in the surrounding time window ($t$(2795) = 2.94, $p = 0.0033$; $\beta = 0.04$, 95% CI = [0.01, 06]), such that greater tone-related VTA activation predicted higher blink count in general (Fig. 4e). Like more local changes in blink count, there was no significant VTA-by-pair type interaction effect on sustained blinking behavior ($t$(2796) = 0.46, $p = 0.64$; $\beta = 0.0058$, 95% CI = [−0.02, 03]), suggesting that momentary VTA activation is coupled with prolonged blinking patterns more generally. Thus, despite differences in their timescales of action, momentary increases in VTA activation were meaningfully embedded within sustained periods of elevated blinking across sequence encoding.

We also found a significant blink-by-pair type interaction effect on temporal distance ratings ($z = 3.17$, $p = 0.0016$; odds ratio = 1.01, 95% CI of odds ratio = [1.00, 1.02]). When analyzing the two pair types separately, we found that blink count significantly predicted temporal distance ratings for boundary-spanning item pairs ($z = 3.16$, $p = 0.0016$; odds ratio = 1.02, 95% CI of odds ratio = [1.01, 1.03]), but not for same-context item pairs ($z = −1.38$, $p = 0.17$; odds ratio = 99, 95% CI of odds ratio = [0.98, 1.00]) (Fig. 4f). This finding suggests that meaningful context shifts predicted stronger coupling between prolonged blinking patterns and the amount of subjective time encoded into long-term memory.

### Identifying which blink periods and stimuli predicted subsequent temporal distortions in memory

The results so far demonstrate that transient blink count in the 1.5 s post-tone-switch did not predict temporal distance ratings between object pairs. Yet, the longer period of blinking between to-be-judged items did predict temporal distance ratings across an event boundary. Because the momentary boundary-related increase in blinking was encompassed within the larger temporal interval between pairs, it is likely the case that blinking in some other time part of the interval (i.e., not specifically within the 1.5 s after the tone switch) was contributing to the observed relationship with time dilation. To explore this nuance of the data, we investigated which specific periods of extended blinking drove the time dilation effect. In brief, we found that the aggregation of tone-evoked blinking during the later portion of the inter-pair interval (i.e., from the tone switch to the appearance of the second to-be-judged item) predicted significant time dilation effects across boundaries ($z = 2.11$, $p = 0.035$; odds ratio = 1.04; 95% CI of odds ratio = [1.00, 1.07]). For more details about these blink analyses and results, see the Supplementary Material and Supplementary Fig. 12.

### Testing the specificity of blink and distance memory measures to the dopaminergic system

Recent findings from this same dataset showed that pupil-linked arousal and locus coeruleus (LC), a key hub of the brain's arousal and noradrenergic system, were associated with impairments in temporal memory[37]. An important open question is whether our current findings on subjective temporal memory and blinking are linked to arousal-related neuromodulation more generally. According to theoretical frameworks of event segmentation[38], both the dopaminergic and noradrenergic systems are well positioned to elicit a global updating signal when event transitions occur. However, it is possible that they contribute to memory separation[39] and blink behavior in different ways. We have also previously shown that LC activation at boundaries relates to more differentiated multivoxel activation patterns in left dentate gyrus (DG), a hippocampal subregion that is important for memory encoding and disambiguating overlapping neural representations[37]. Much work shows that dopaminergic modulation of hippocampal processing also plays an essential role in structuring and encoding episodic memories, making it another candidate input signal for memory separation at its target regions[12].

To test the specificity of dopaminergic effects on temporal and hippocampal processing, we ran several mixed effects modeling analyses that complemented those in Clewett et al. [37], which had focused only on LC-related effects. We then also performed follow-up modeling to test whether relationships between VTA activation, eye-tracking measures (pupil dilation and blinking), hippocampal pattern stability, and temporal memory differed from those of the LC (see Supplementary Material for details on methods and results). In brief, we found that LC activation was more strongly coupled with impairments in temporal order memory, an objective behavioral index of memory separation, than VTA activation. However, VTA activation was not

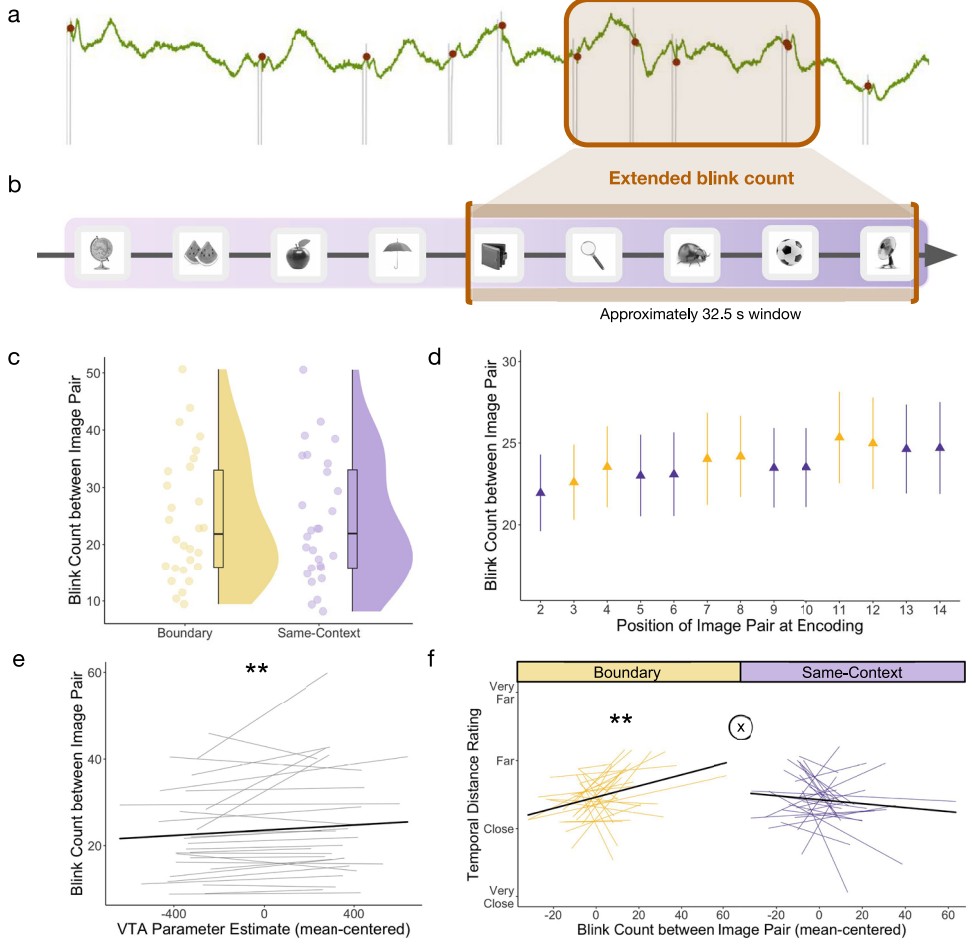

**Fig. 4 | Event boundary-evoked VTA activation predicted greater sustained blinking between to-be-tested item pairs, and this blink pattern predicted later time dilation specifically between boundary-spanning item pairs in memory.**
**a** Example pupil timeseries for a subset of an encoding block. **b** Schematic of blink window of interest during event sequence encoding. Extended blink count was captured during each approximately 32.5 s window between the onset and offset of to-be-tested item pairs (bracketed in orange). **c** Overall comparison between mean blink count between to-be-tested images spanning an event boundary (yellow) or from the same context (purple). Each dot represents averaged data from one participant (*n* = 28 participants). For box plots, boxes are defined by 25th percentile, median, and 75th percentile. Whiskers are defined by minimum and maximum, excluding outliers. **d** Mean blink count during these longer time windows, plotted by item pair position at encoding (14 tested pairs total; first pair excluded). Yellow triangles represent boundary-spanning item pairs, while purple triangles represent same-context item pairs. In order of pair presentation at encoding, the boundary-spanning pairs were the 3rd, 4th, 7th, 8th, 11th, and 12th item pairs. Error bars represent SEM. *n* = 28 participants. **e** Participant-level trendlines plotting the relationship between trial-level VTA parameter estimates (mean-centered) and more

sustained blinking patterns between to-be-tested item pairs. The dark, bold line represents the average linear trend across participants. Statistical significance refers to results from trial-level linear mixed effects model (*p* = 0.0033).
**f** Participant-level trendlines plotting the relationship between extended blink count (mean-centered) and temporal distance ratings for boundary-spanning pairs (yellow; left) and same-context pairs (purple; right). For modeling, blink count was mean-centered for each condition separately. The dark, bold line represents the average trendline across participants, also for each condition separately. Statistical significance refers to results from trial-level cumulative link models (boundary: *p* = 0.0016. The circled X symbol represents a significant condition-by-blink count interaction effect on temporal distance ratings with *p* = 0.0016. **p* < 0.01. Source data are provided as a Source Data file. The images in this figure are credited to: Alex Staroseltsev/Shutterstock.com; GalapagosPhoto/Shutterstock.com; Bilda-gentur Zoonar GmbH/Shutterstock.com; Sergiy1975/Shutterstock.com; Anton Starikov/Shutterstock.com; James.Pintar/Shutterstock.com; Preto Perola/Shutter-stock.com; tanadtha lomakul/Shutterstock.com; Aleksey Sagitov/Shutter-stock.com; and iStock.com/ramzihachicho.

significantly more coupled with distance ratings than LC activation, so we cannot make strong claims about specificity. On the other hand, blinking was more strongly coupled with VTA activation than LC activation, and more strongly coupled with distance memory than impairments in order memory.

When probing links to the stability of hippocampal multivoxel patterns across time, LC activation was significantly more coupled with pattern differentiation in left DG compared to VTA activation. This suggests that noradrenergic processes may have a stronger influence over ongoing event representations supported by the hippocampus. Finally, VTA activation was not significantly correlated with different features of boundary-evoked pupil dilation[37]. For more details

concerning all these analyses and results, see Supplementary Material and Supplementary Figs. 6–11.

Taken together, our results suggest that extended blinking behavior is indeed closely tied to dopaminergic activity as opposed to arousal processes more generally. However, they suggest that the noradrenergic system, and not the dopaminergic system, may modulate memory separation by influencing objective aspects of temporal memory and differentiating hippocampal representations.

## Discussion
Segmenting continuous experience into distinct events helps structure memories in ways that support adaptive behavior. One way this

structure might be achieved is through the subjective stretching of remembered time across an abrupt change in context[4,5,36,40,41], such as a shift in location, goal, or mood. Although these subjective distortions can belie the actual passage of time, they may serve a functional role in delineating boundaries between events in both perception and memory. In the current neuroimaging study, we replicated this time dilation effect in memory and found that context shifts, or event boundaries, elicited transient increases in ventral tegmental area (VTA) activation—a key hub of the dopamine system—and blinking. VTA activation was associated with both momentary and sustained increases in blink count, supporting a putative link between blink behavior and dopaminergic activity. Finally, stronger VTA responses and elevated blink counts across boundaries predicted greater time dilation between item pairs spanning those same boundaries. Together, these converging results suggest that dopaminergic signaling may contribute to the mnemonic separation of distinct events by expanding the temporal gap between them.

Our findings help bridge separate lines of research linking dopamine to episodic memory and temporal processing. We first showed that context shifts predicted stronger activation of the VTA, consistent with prior work showing that dopamine flags novel and salient stimuli[11,12]. Context shifts also predicted brief increases in blinking, aligning with studies showing that blinks tend to occur near event transitions, including punctuation marks in text passages or breakpoints in everyday conversations[31,32,34] (but also see Nakano et al. [30] for an alternative interpretation). Specifically, we found that these blinks occurred after boundary tones, not before them. This timing is consistent with evidence that blinking tends to be delayed when a motor response is required[25], such as a button press response to an orienting question (Is this object larger or smaller than a standard shoebox?). Moreover, VTA responses to tones generally predicted transient increases in blinking. This finding aligns with recent evidence that discrete blinks are associated with fMRI activation in dopaminergic regions, including the VTA and substantia nigra[25]. Overall, these results support the idea that VTA activation and blinks mark transition points in experience.

Why might dopaminergic processes be engaged at event boundaries? One possibility is that dopamine facilitates the rapid updating of event models during sudden or unexpected changes in the environment. Supporting this view, prior neuroimaging work shows that dopaminergic brain regions – such as the substantia nigra and basal ganglia – are activated when participants make predictions about the next few seconds of videos depicting everyday activities (e.g., gardening). When these intervals contain an event boundary, participants tend to make less accurate predictions[14]. Dopamine may signal high prediction error at event boundaries, indicating that a new mental representation of the world is needed to represent unfolding events more accurately[14,24]. It has also been shown that VTA activation relates to moments of surprise in more naturalistic settings, such as watching videos of basketball games[13]. However, given that the structure of our task was largely predictable, we speculate that it is unlikely that prediction errors and surprise accounted for boundary-related patterns of VTA activation in the current sequence paradigm.

An alternative, more plausible explanation for boundary-evoked dopaminergic activity is that it signals contextual novelty during dynamic experience. It has been theorized that the VTA releases dopamine into the hippocampus in response to common novelty—that is, stimuli that are novel under current conditions but not completely new[11]. Following this logic in the current study, each context change was relatively novel within its local temporal context (i.e., block), because the tone switches occurred infrequently and introduced a pitch that was unique to a given block. Thus, dopamine might signal a kind of common novelty at these transitions within sequences, marking the end of one meaningful event and the start of the next.

Consistent with our key hypothesis, we also found that higher boundary-related VTA activation predicted greater time dilation effects in subsequent memory, with individuals remembering item pairs as being encountered farther apart in time if they spanned a boundary. This finding aligns with prior research showing that dopaminergic processing relates to changes in internal representations of time. For example, studies in humans have found that encountering salient oddball stimuli, linked to dopamine signaling[42], can lead individuals to overestimate the duration of those stimuli[43–45]. However, we note that this effect of VTA activation on time dilation did not differ between conditions, so we cannot draw strong conclusions about specificity. It is possible that this VTA-memory relationship reflects processes beyond boundary-specific signaling, such as general salience or updating mechanisms. Nevertheless, because the coupling between boundary-evoked VTA activation and temporal memory emerged at the trial level, our results suggest that transient dopaminergic responses at event boundaries might meaningfully shape how temporal distance is encoded. We also note that interpreting VTA-memory relationships for same-context pairs is inherently constrained by choosing a single tone to index VTA activation: the reference tone that was position-matched to the tone switch between boundary-spanning pairs. It remains possible that there are activity patterns at other timepoints within this same-context pair interval that could yield a different pattern of results.

The time dilation effect we observed aligns with existing conceptual frameworks of temporal processing, including the dopamine clock hypothesis and contextual-change hypothesis of remembered duration. According to these models, both dopamine[19,20] and the number of contextual changes stored in memory[46,47] serve to expand internal representations of time. One intriguing possibility is that boundary-induced VTA activation and dopamine release boost encoding of local context changes, constructing more memories of salient experiences. As a result, there is a greater amount of episodic information and details available at retrieval, leading to overestimations of retrospective time. Future work could test this idea by examining whether boundary-induced VTA activation relates to increases in local source encoding processes, as boundaries have been shown to reliably enhance source memory for concurrent information[5,6,36]. We suggest that these dopamine-mediated temporal distortions may occur via modulation of hippocampal activity, given its well-known role in supporting the storage and retrieval of temporal and source information[48].

On a larger timescale, we observed that prolonged periods of blinking across boundaries predicted remembered time. This relationship was stronger for periods spanning contextual change versus contextual stability, suggesting that boundary-specific features like contextual novelty and task relevance might be critical for shaping memories of time in a meaningful way. Indeed, our analytical approach to the blink data was notable in that prior eye-tracking studies have either measured discrete blinks after individual stimuli[35] or longer periods of spontaneous blinking offline or during rest. For example, spontaneous blink rate has been linked to dopaminergic processes in non-human primates and humans with neuropsychiatric disorders[28]. Resting-state measures of spontaneous blinking are likely noisy, given that blinking is affected by both cognitive and non-cognitive factors, such as fatigue. The current study assessed blinking during a behaviorally relevant time window in memory encoding, perhaps leading to a stronger and more meaningful coupling between blinking and dopaminergic activity. One possibility, then, is that blinking may better track dopamine dynamics online and over longer periods of time, potentially reconciling some of the mixed findings in the blink-dopamine literature[25,28,49–52].

There are several limitations in the current study that warrant consideration. First, fMRI cannot establish a causal link between dopaminergic processing and time dilation in memory, nor can it

directly assess neurotransmitter activity. Second, while we used a highly specific, data-driven algorithm to locate blink artifacts, examining electrical activity near the eye offers a more direct method of detecting blinks[53]. Third, there are mixed findings concerning the existence or strength of an association between blinking and dopaminergic processing. Some studies challenge the relationship between certain measures, such as striatal dopamine synthesis capacity, and blinking[54–56]. Additional factors that could influence blink behavior include the activity of other neuromodulators like serotonin[57], as well as inattentiveness or fatigue[58]. However, because event boundaries typically elicit an increase in attention[59], it is unlikely that the effects we observed were solely or predominantly driven by fatigue or time-on-task effects[58,60]. Future studies should examine how relations between blinking and cognitive processing may be altered under different conditions, including whether blinking is measured online or during rest.

In summary, our findings revealed that different indirect markers of dopaminergic processing are related to time dilation effects in memory. We speculate that these subjective time distortions are highly adaptive, because they might serve to increase the representational distance between adjacent episodes in memory. These results are an important first step towards unraveling the complex role of dopamine activity in shaping everyday memory organization. Still, additional techniques are needed to assay neuromodulatory activity more directly and at a higher temporal resolution[61]. Ultimately, these findings will promote a deeper understanding of how internal representations of time mold the complexity of our experience into a meaningful record of past events.

## Methods

### Participants
Thirty-six participants were recruited from the New York University (NYU) Psychology Subject Pool and the local community. Sample size was estimated using a G*Power 3.1 power analysis (alpha = 0.05, power = 0.80, $d$ = 0.80), based on the pooled temporal memory performance from a very similar event boundary experiment[36]. This power analysis yielded a required sample size of 29 participants. To account for attrition, we recruited 36 participants. Participants were required to have normal or corrected-to-normal vision and hearing, to have no metal in their body, and to not be taking beta-blockers or psychoactive medications. All participants provided written informed consent approved by the NYU Institutional Review Board and received monetary compensation for participation.

Four participants were fully excluded from analyses (reasons included falling asleep in the scanner and audio malfunction). Therefore, our final sample size was 32 participants (20 F; $M_{age}$ = 22 years, $SD_{age}$ = 2.7 years). Of this sample, 15 participants identified as Asian, 2 as Black/African American, 4 as more than one race, and 11 as White.

Of these 32 participants, five requested to leave the scanner early and thus did not complete all 10 blocks of the task ($n$ = 2 completed 9 blocks, $n$ = 1 completed 8 blocks, $n$ = 2 completed 7 blocks). Finally, four participants' eye tracking data was excluded due to equipment malfunction or poor quality, leaving 28 participants with usable data for all analyses that included blink data.

Sex differences were not analyzed or included as covariates in any analyses, because they were not central to our predictions and there were relatively small sample sizes of each type.

### Materials
Participants viewed 512 grayscale images of everyday objects[62,63], which were resized to 300 × 300 pixels, placed on a gray background, and luminance-normalized using the MATLAB SHINE toolbox to control for the effect of brightness on pupil behavior. To embed these images within distinct auditory contexts, participants were presented with 1 s pure tones (created with Audacity; https://www.audacityteam. org/) in either the left or right ear. The six tone frequencies used ranged from 500-1000 Hz in 100 Hz intervals, were perceptually discriminable, and were sufficiently arousing to facilitate engagement in the task.

### Code and data accessibility
The code, experiment materials, and data for this study are publicly available on the OSF account of E.M. (osf.io/yt6hm).

### Overview of experimental design
This experiment took place over two days. On the first day, participants completed several questionnaires and a practice task block before completing the full event sequence task in the MRI scanner. The MRI session took approximately 3 h, with 2.5 h of scanning. On the second day, approximately 24 h later, participants returned to perform a surprise item recognition memory test (not described in the current paper).

### Event sequence encoding task
While in the MRI scanner, participants completed a modified version of an event sequence encoding task that has been shown to elicit reliable event segmentation effects in memory, as indexed by significant time dilation in retrospective distance ratings[36] (Fig. 1). Our goal was to use these boundary-induced subjective temporal distortions to operationalize whether memories had become separated at context shifts. That is, if item pairs spanning a boundary were remembered as appearing farther apart than item pairs from the same context—despite being the same objective distance apart during encoding—then they were more likely to have become encoded as distinct episodic memories.

In this encoding-retrieval block design, participants were presented with sequences of 32 object images for 2.5 s each. For each image, participants made a button press to respond to a simple orienting question (Is this object larger or smaller than a standard shoebox?). Additionally, participants were asked to form their own mental narrative of the image sequence to encourage associative encoding[64]. A jittered ISI (3, 5, or 7 s fixation cross) separated each image presentation to optimize deconvolution of the blood-oxygen level dependent (BOLD) signal during the fMRI analyses.

Each item sequence was divided into four auditory events that contained 8 images each. Events were defined by a stable auditory context, in which 1 s pure tones played either in participant's left or right ear only, halfway through each jittered ISI. The laterality of the tone also cued participants to use either their left or right hand to make button responses for the object size judgments (left ear = left hand). Thus, the tones not only provided sensory information to elicit perception of stable mental events, but were also task-relevant. After the final image in an 8-item event, the tone abruptly switched sides to the opposite ear and changed pitch. Participants were also instructed to switch the hand used to make button responses. This salient change constituted an event boundary in the sequence, marking the end of one event and the beginning of the next. The new context then remained the same for the rest of the 8-item event.

Each item sequence contained three tone switches total, resulting in four stable auditory events. There were 10 encoding-retrieval blocks total across the experiment. The starting ear/hand was counterbalanced across blocks. Pitch changes were pseudorandomized, such that the same frequency (e.g., 500 Hz) was not heard more than once in any given encoding block. To separate the encoding and retrieval tasks and to reduce any recency effects in memory, participants completed a brief arrow detection task (45 s) after each encoding block. Participants were presented with 0.5 s left-facing (<) or right-facing (>) arrows in the center of the screen, separated by 0.5 s ISIs of fixation. They were instructed to indicate which direction the arrow was facing via button press as quickly as possible.

## Temporal distance memory test

At the end of each encoding block, participants completed two temporal memory tests: the first for temporal order and the second for temporal distance. Given that our primary focus was on subjective distortions in distance memory, the results from the temporal order memory test are reported elsewhere[37].

In the temporal distance memory test, 14 item pairs from the encoding sequence were presented for a fixed duration of 5 s each. Importantly, each of these item pairs had been separated by exactly three images during encoding. Because the objective distance between to-be-tested pairs was pseudorandomized to always be equivalent (approximately 32.5 s from onset of the first image to the offset of its pairmate), we were able to measure subjective temporal distortions in memory across conditions.

Accordingly, participants were asked to judge how far apart in time two images appeared. There were four options: very close, close, far, and very far. To replicate previous effects of event boundaries on temporal distance memory, we compared ratings for two types of item pairs: same-context pairs (8 trials per block) and boundary-spanning pairs (6 trials per block). Same-context pairs contained images from the same context during encoding, while boundary-spanning pairs contained images that spanned a change in context (i.e., task-relevant tone switch) during encoding.

We excluded the first same-context item pair from all analyses, because this pair contained the first image in each block. As such, it likely constituted a task-irrelevant event boundary and would therefore produce different behavioral effects than the other same-context pairs. Due to a programming error, one of the boundary pairs from each list also contained an incorrect item. Data for these specific pairs (appearing once per block across 23 participants) were excluded from the analyses. Finally, one block was excluded for 9 participants due to a timing error.

## fMRI acquisition and preprocessing

**fMRI/MRI data acquisition.** All neuroimaging data were acquired with a 3 T Siemens Magnetom PRISMA scanner using a 64-channel matrix head coil. First, participants underwent a high-resolution MPRAGE T1-weighted anatomical scan (slices = 240 sagittal; TR = 2300 ms; TE = 2.32 ms; TI = 900 ms; FOV = 230 mm; voxel in-plane resolution = 0.9 mm²; slice thickness = 0.9 mm; flip angle = 6°; bandwidth = 200 Hz/Px; GRAPPA with acceleration factor = 2; scan duration: 5 m 21 s).

Next, participants underwent a T2-weighted functional scan (slices = 240 sagittal; TR = 3200 ms; TE = 564 ms; FOV = 230 mm; voxel in-plane resolution = 0.9 mm²; slice thickness = 0.9 mm; flip angle = 6°; bandwidth = 200 Hz/Px; GRAPPA with acceleration factor = 2; scan duration: 3 m 7 s). The task audio was also calibrated during this scan to ensure the tones were audible above scanner noise and could be discriminated from one another. Participants could make button presses to request changes in tone volume. Additionally, we collected two fieldmap scans to assist with functional imaging unwarping (1 in anterior-posterior (AP) phase encoding direction and 1 in posterior-anterior (PA) phase encoding direction). Afterwards, we collected a short fast spin echo (FSE) sequence MRI scan that enables visualization of the locus coeruleus (LC). Details are reported elsewhere[37].

After the structural scans were collected, participants underwent separate functional imaging for each of the 10 encoding blocks and the 10 retrieval blocks. Functional scans were collected using a whole-brain T2*-weighted multiband echo planar imaging (EPI) sequence (128 volumes per encoding block; TR = 2000ms; TE = 28.6 ms, voxel in-plane resolution = 1.5 × 1.5 mm²; slice thickness = 1 mm with no gap; flip angle = 75°, FOV = 204 mm × 204 mm; 136 × 136 matrix; phase encoding direction: anterior-posterior; GRAPPA factor = 2; multiband acceleration factor = 2). In each volume, 58 slices were tilted −20° of the anterior commissure-posterior commissure line and were collected in an interleaved order. A single-band reference image was collected for each run of the task, but was not included during preprocessing.

**fMRI preprocessing.** Image preprocessing was performed using FSL Version 6.00 (FMRIB's Software Library, www.fmrib.ox.ac.uk/fsl). Functional images were preprocessed using the following steps: removal of non-brain tissue using BET; B0 unwarping using fieldmap images; grand-mean intensity normalization of the 4D data set by a single multiplicative factor; and application of a high-pass temporal filter of 100 s. No spatial smoothing was applied due to the small size of the VTA and to preserve spatial specificity. Motion correction was performed using the MCFLIRT tool, which produced six motion nuisance regressors. Additionally, fsl_motion_outliers was used to identify volumes with extreme head movements, or frame displacements, using the DVARS option. Both this matrix of outlier volumes and the six motion regressors were as modeled as covariates in the subsequent GLM analyses. Entire blocks with excessive head motion overall (conservatively defined as mean frame displacement > 1 mm) were excluded from analysis, resulting in the removal of one block each from three participants. Each participant's denoised mean functional volume was co-registered to their T1-weighted high-resolution anatomical image using brain-based registration (BBR). Anatomical images were then co-registered to the 2 mm isotropic MNI-152 standard-space brain using an affine registration with 12 degrees of freedom.

**Physiological denoising.** Eight separate physiological nuisance signal regressors were extracted for the subsequent GLM analyses. First, FSL FAST was used to decompose each participant's high-resolution anatomical images into probabilistic tissue masks for white matter (WM), gray matter (GM), and cerebrospinal fluid (CSF). The CSF and WM masks were thresholded at 75% tissue-type probability to increase their spatial specificity and reduce potential overlap. Following a similar approach to Bartoň et al. [65], we defined eight 4 mm spheres in representative regions of WM and CSF (four of each type; for exact coordinates, see Bartoň et al. [65]). The eight spheres and WM and CSF anatomical masks were then transformed into each participant or block's native functional space and merged to further increase their spatial specificity. Nuisance timeseries for each of the four WM and four CSF merged masks were then extracted from each block's preprocessed functional data and modeled as nuisance regressors in the GLMs.

**VTA region-of-interest (ROI) definition.** To create an anatomical mask of the VTA, we used a publicly available probabilistic atlas that was originally created using individual hand-drawn ROIs[66]. This standard-space VTA mask was thresholded at 75% probability to increase its spatial specificity and then registered into each participant's functional run (i.e., block) of the encoding task.

**Hippocampal subfield segmentation and quality control.** We used Freesurfer Version 6.0 (http://surfer.nmr.mgh.harvard.edu/) to segment bilateral hippocampal subfields CA2/3, DG, and CA1 from each participant's high-resolution anatomical scan. To quality control this segmentation, we used guidelines designed for the Enhancing Neuro Imaging Genetics through Meta-Analysis (ENIGMA) consortium (https://enigma.ini.usc.edu/ongoing/enigma-hippocampal-subfields/). Details are reported elsewhere[37].

## fMRI analyses

**Generalized linear modeling (GLM) analyses and acquisition of single-trial activation estimates of VTA activation.** To estimate VTA activation at the trial level, we conducted Least Squares Separate (LSS) GLM analyses on the unsmoothed functional data from the sequence encoding task. This modeling approach generates unique activation estimates for each stimulus (i.e., tone or image) from the task.

Importantly, the repetition or switching of tones carried the critical information about the stability or change in the surrounding context. Thus, using LSS GLM enabled us to isolate the distinct effects of context-relevant tones on brain activation.

Each LSS-GLM contained a total of 64 task-related regressors for each block of the encoding task, because there were 32 tones and 32 images in each block. Each tone was modeled as a 1 s stick function and each image was modeled as 2.5 s stick function. Both regressors were convolved with a dual-gamma hemodynamic response function (HRF). A total of 64 separate LSS-GLM analyses were conducted for each block of the ten blocks of the task, where one stimulus (image or tone) served as the regressor of interest and all other trials were modeled as a separate regressor. Each LSS-GLM thereby resulted in a unique activation estimate (i.e., beta map) across the whole brain for each stimulus[67,68]. To control for noise, a total of 14 nuisance regressors (4 WM, 4 CSF, and 6 motion regressors) were included in each GLM, along with individual nuisance regressors for trials with extreme head movements.

A VTA ROI analysis was performed on the resulting beta maps. The standard-space VTA anatomical mask was transformed into each participant's native run-space for each encoding block and used to extract single-trial VTA parameter estimates (a measure of activation) for each of the 32 tone trials (3 boundary tones, or switches, and 29 same-context tones, or repeats). We excluded noisy datapoints using boxplot outlier removal by participant, resulting in the removal of approximately 3% of the tone-related VTA activation trials across the entire dataset.

### Eye-tracking methods

**Eye-tracking.** Pupil diameter was measured continuously at 250 Hz during the event sequence task using an infrared EyeLink 1000 eye-tracker system (SR Research, Ontario, Canada). Raw pupil data, segmented by block, were preprocessed using ET-remove-artifacts, a publicly available MATLAB program (https://github.com/EmotionCognitionLab/ET-remove-artifacts). Although this software is typically used to clean pupil timeseries by interpolating over blink events and other artifacts[37], our primary objective was to identify blink event timestamps and quantify their frequency within different periods of interest.

ET-remove-artifacts located blink events by identifying rapid changes in pupil size, or pupil velocity, following the approach described in Mathôt et al. [69]. The velocity timeseries was computed by applying MATLAB's finite impulse response (FIR) differentiator filter on the raw pupil size timecourse. This method provides a robust estimate of instantaneous rate of change while minimizing noise amplification. For our dataset, we selected FIR filter parameters (Filter Order = 4, Passband Frequency = 10, and Stopband Frequency = 12) to produce a smooth velocity timeseries with trough-and-peak profiles that were identifiable and specific to blink events rather than noise-related data loss.

The algorithm then used MATLAB's "findpeaks" function to locate peaks and troughs in the pupil velocity timecourse. Blink profiles were distinctly identifiable as a contiguous trough followed by a peak in the velocity timeseries. To achieve a high degree of specificity to blink events, we set the Peak and Trough Threshold Factor and Trough Threshold Factor at 8 standard deviations of the velocity timeseries. This threshold is higher than those typically used for cleaning artifacts from pupil timeseries[37], ensuring that only substantial blink-related troughs and peaks were counted as blink events rather than small and transient velocity changes attributable to other artifacts.

To generate a preprocessed pupil timecourse, the algorithm applied a linear interpolation across identified blink intervals. Artifact intervals exceeding 2 seconds were automatically imputed with NaN (missing data indicator).

Alongside the blink estimates, we also computed pupil dilation responses to both boundary tones and same-context tones. A temporal principal component analysis (PCA) was applied to these tone-evoked pupil responses to dissociate distinct autonomic and functional components of pupil-related arousal. Details about pupil analyses and methods are reported elsewhere[37], because the goal of this manuscript was to focus on the specificity of the relationship between brainstem nuclei activation and blinking measures of neuromodulation. As mentioned previously, four participants were excluded from eye tracking analyses due to system malfunction or poor overall eye tracking quality, resulting in a total of 28 participants for all eye tracking-related analyses.

**Computing local blink counts.** Breakpoints in continuous experience have been shown to elicit a momentary increase in blinking[31,34]. Further, blink behavior has also been putatively linked to dopaminergic processes[28], suggesting that blink behavior may offer a window into neuromodulatory processes that facilitate memory separation at boundaries. To test this idea, we used MATLAB Version R2022b to tabulate local blink count in a 1.5 s interval immediately after the onset of each tone. The size of this interval was chosen because the smallest ISI in the current paradigm was 3 s – that is, 1.5 s on either side of the tone onset. In this way, we were able to capture tone-induced blinks during a brief period that was uncontaminated by the preceding or ensuing images on any trial. To acquire highly conservative estimates of blink behavior that were not confounded by noise or data loss, blink intervals with more than 25% invalid samples were excluded from analyses. As with the VTA data, we also cleaned the trial-level blink responses using boxplot outlier detection within each participant. After these exclusions, approximately 78% of local blink intervals remained per participant, on average.

**Computing sustained blink counts between to-be-tested item pairs.** We also zoomed out to examine blink patterns across longer, behaviorally relevant windows of encoding to assess more sustained dopaminergic processes. We again used MATLAB Version R2022b to calculate blink count for the approximately 32.5 s window between each to-be-tested pair, from the onset of the first item to the offset of its pairmate (encountered four images later). We conducted the same noise-related and by-participant boxplot outlier removal procedure as before, which left approximately 79% of blink windows per participant, on average.

**Testing linear relations between the VTA, temporal memory, and blink patterns.** To assess trial-level relationships between our key variables, we used linear mixed-effects and cumulative link modeling. Linear mixed-effects modeling was used when the outcome of interest was continuous (i.e., trial-level VTA parameter estimates, post-tone blink count, and extended blink count). Fixed effect predictors in these models were categorical (i.e., tone type: boundary tone, same-context tone) and/or continuous (i.e., trial-level VTA parameter estimates).

Cumulative link models are the most common type of ordinal regression model, using a similar frequentist approach to linear mixed-effects models[70,71]. Thus, cumulative link modeling was used when the outcome of interest was ordinal (i.e., trial-level temporal distance memory ratings). Distance memory ratings were ordered from the smallest to largest rating: very close, close, far, very far. Fixed effect predictors in these models were categorical (i.e., pair type: boundary-spanning pair, same-context pair) and/or continuous (i.e., trial-level parameter estimates, post-tone blink count, and extended blink count). In the models with trial-level VTA parameter estimates as a fixed-effect predictor of distance memory, we included an additional categorical variable for Pair Position during encoding to account for potential list position effects. Including this

predictor improved model fit ($ps < 0.01$). This predictor was also included in associated modeling analyses in the Supplementary Material (see Supplementary Figs. 6-7).

To meaningfully relate local measures (i.e., VTA parameter estimates and post-tone blink counts) to measures capturing a wider temporal window (i.e., extended blink count and distance memory ratings), we extracted the value associated with one tone between each to-be-tested item pair. For details on the exact positions of the selected tones and their position-matching across boundary and same-context pairs, see Supplementary Figs. 1-2.

Analyses were conducted in RStudio (Version 2024.2.29)[72], with the main modeling using the lme4[73] and ordinal packages[74]. Each model included random intercepts for participant ID. Continuous predictors were mean-centered by participant to reduce the influence of individual differences[75]. This mean centering was performed separately for each model. Continuous predictors were also $z$-scored as necessary to improve model convergence. When the outcome of interest was continuous, standardized parameters were obtained by fitting the model on a standardized version of the dataset. 95% confidence intervals were computed using a Wald $z$-distribution approximation. When the outcome of interest was ordinal, effect sizes were computed as odds ratios (with 95% confidence intervals).

### Reporting summary
Further information on research design is available in the Nature Portfolio Reporting Summary linked to this article.

## Data availability
Source data are provided with this paper. The processed data are also publicly available on the OSF account of Erin Morrow at the following link: osf.io/yt6hm. Source data are provided with this paper.

## Code availability
The code for this experiment is publicly available on the OSF account of Erin Morrow at the following link: osf.io/yt6hm[76].

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

## Acknowledgements

This project was funded by federal NIH grant R01 MH074692 to Lila Davachi and by fellowships on federal NIH grant F32 MH114536 to D.C. We thank Jaime Castrellon, Alexandra Cohen, and Jacinda Taggett for very helpful feedback on earlier versions of this manuscript. We thank Emily Morrow for assistance with the figures in this manuscript.

## Author contributions

D.C. conceptualized and designed the experiment. D.C. collected data. E.M., D.C., and R.H. analyzed the data. E.M., D.C., and R.H. wrote the manuscript.

## Competing interests

The authors declare no competing interests.
