## [Transparent Peer Review file · Nature Communications]

Dopaminergic processes predict temporal distortions in event memory

Corresponding Author: Dr David Clewett

Version 0:

Reviewer comments:

Reviewer #1

(Remarks to the Author)

In this study authors explored the blink rate changes, VTA activity and short-term memory associative chunking (event boundaries and chunking intervals). While I found the study interesting, methods and approach valid, and the citations reliable, I have some concerns in experimental design and conclusions drawn from it. So I am not sure I would accept the paper for publication in its current state. Maybe the authors can help me to see the paper in a better light.

- Major issue is primary-recency effects: in short-term memory research, it is a well-known fact that the middle items in a sequence of item-lists are most likely to be forgotten. When a subject is asked a pair of objects which are 5-items apart among 8-item lists of chunks, the likelihood that this pair would be where one item falls on the middle section and the other one on the prior or recent section in the same chunk group, while it always falls as primary-recent items association between the separate chunks/events. When people are asked to recall/recognise item-pairs, question was 'how far they are apart from each other'. It is possible that recognized items are more likely to be associated and perceived as closer to each other, while fanned-out items might be perceived as more distant. I need more memory literature in the intro including Baddeley's theories.
- Authors can benefit from P300 ERP literature related to contextual updating and its potential relationship with the VTA and ACC. I also encourage authors to report ACC region.
- Authors did not use false item-pairs in which items not memorized are shown, at all as a control condition. Nor did they use different distance metric (maybe in some cases 7-items distance) to test against the effect of recency.
- It would be better to use longer time window to count blinks. I understand that they used 1.5s window because it was the shortest epoch, however I would use all the time segment after the beep till the first image of the next chunk was shown. They can split long and short windows and report separately.
- Authors mentioned LC. However, this area is more likely to be related to pupil size. If they are not going to report pupil size I would ask them to take it out of the paper. Or include pupil size.
- There are a few outlier subjects who always pressed 'very far' or very close'. I do not think these subjects performed the task at all and should be taken out. You can also look at the box-test results and it is very likely that these subjects always pressed big or always small (ehre are the accuracy reports?) I understand that the sample size is already reduced but this will be more appropriate to do.
- Demiral 2023 has interesting supplementary material; looking at a gambling task and explored button presses and blink timing. This might be helpful to notice how blinks wait for wrap-up of the motor action.
- Authors used multi-band imaging; I wonder whether collecting single band reference scan would improve preprocessing (authors did not mention they collected it).

Overall, I appreciate the work behind this study and that authors kindly provided open-access for their pipelines and data. If they can provide satisfactory answer to my comments and questions I can recommend their paper for publication.

(Remarks on code availability)

I was able to see code under <https://osf.io/8y7hr/files/>.

Reviewer #2

(Remarks to the Author)

Review of Morrow et al., "Dopaminergic processes predict temporal distortions in memory"

This paper explores the role of the dopaminergic (DA) system in driving time dilation effects that are commonly observed at event boundaries. To study these effects, the authors use a variant of the Ezzyat-DuBrow-Davachi event segmentation paradigm, where changes in the pitch of tones presented in between object pictures are used to trigger event boundaries. The paper reports several important findings: tone switches activate VTA; the magnitude of the VTA response at tone switches predicts the level of behavioral time dilation; tone switches predict increased blinks; VTA activation predicts blinking in general; and higher blink rates between to-be-judged items predict greater time dilation, but only for boundary-spanning pairs. Overall, I think this is an exciting package of results and I am very enthusiastic about the paper – but I do see some issues with the present version of the paper that I'd like the authors to address.

The first set of issues relates to how specific these results are to the DA system. Other results from this same study, focused on how the noradrenergic (NE) system is engaged at event boundaries, were written up in a separate manuscript that is presently in press at Neuron. This paper and the Neuron paper focus on almost completely distinct sets of dependent variables. The present paper focuses on VTA activation to tone switches, temporal distance judgments, and blink rate; the Neuron paper focused on LC activation to tone switches (indicative of the engagement of the NE system), temporal order judgments ("which of these two items came first"), pupil diameter, and hippocampal pattern similarity. The present paper, as noted above, found that VTA activation at boundaries relates to temporal distance judgments and blink rate. The Neuron paper found that LC activation at boundaries relates to temporal order judgments, pupil diameter, and hippocampal pattern similarity. Given these results, it is natural to wonder whether the DVs that relate to VTA activation also relate to LC activation, and vice-versa. The present paper addresses this in a brief section on page 11 where the authors show that LC activation is not significantly related to temporal distance ratings or blink rate, and that VTA activation is not related to temporal order judgments. While this is suggestive of a double dissociation between the two systems, there are some missing comparisons (outlined below), and also the authors do not report any exact statistics or effect sizes and they don't show any results plots comparing the effects. To more firmly establish this double dissociation, I suggest that the authors run the following analyses:

It would be useful to show, statistically, that the relationship between VTA responses to tone switches and temporal distance is larger than the relationship between LC responses to tone switches and temporal distance (presently, the authors only show that the latter relationship is nonsignificant).

It would be useful to show, statistically, that the relationship between LC responses to tone switches and temporal order is larger than the relationship between VTA responses to tone switches and temporal order (presently, the authors only show that the latter relationship is nonsignificant).

It would be useful to show, statistically, that the relationship between VTA responses to tones and blink rate is larger than the relationship between LC responses to tones and blink rate (presently, the authors only show that the latter relationship is nonsignificant).

It would be useful to show, statistically, that the relationship between LC responses to tone switches and pupil diameter is larger than the relationship between VTA responses to tone switches and pupil diameter.

It would be useful to show, statistically, that the relationship between LC responses to tone switches and DG pattern similarity is larger than the relationship between VTA responses to tone switches and DG pattern similarity.

In the methods section, the authors should either refer to the Neuron paper for methods relating to LC activation and pupil diameter measurements, or else they should copy those methods into the present paper.

The second set of issues relates to the blink rate results. The authors show that blink rate in the 1.5 sec post-tone-switch did not predict temporal distance ratings, and they also show that average blink rate between to-be-judged items predicts temporal distance ratings, but only when there's an event boundary. Since blink rate within-1.5-sec-post-tone-switch is factored into the average blink rate between the to-be-judged items, this means that blinks in some other time interval (i.e., not within 1.5 sec after the tone switch) are contributing to the observed relationship to time dilation. It seems like the authors could do more to nail down which parts of the interval between the to-be-judged items are driving the effect. For example, it seems unlikely that blinks in the part of the interval before the tone switch are driving the effect, and relatively more likely that blinks after the tone switch (i.e., between the tone switch and the appearance of the second to-be-judged item) are driving the effect – it would be straightforward to run this analysis. And if blinks after the tone switch are driving the effect, but blinks in the 1.5 sec immediately after the tone switch are not driving the effect, this suggests that blinks even later post-tone-switch are driving the effect. Are these blinks coming in response to the items? or in response to the later tones? In general, how often do participants blink outside of the 1.5 sec window following tones? Gaining a better understanding of which parts of the interval are driving the predictive relationship to time dilation would add to the overall story.

MINOR ISSUES

p. 12: "elevated blink rates at event boundaries predicted greater time dilation"... this is not an accurate description of the finding

p. 12: "VTA responses to boundary tones... predicted transient increases in blinking" – this is a bit misleading, as VTA

responses to non-boundary tones also predicted transient increases in blinking (fig. 3e)

For analyses relating average blink rate between the to-be-judged items and tone-related VTA activation, when was tone-related VTA activation measured (Figure 4e)? My understanding was that both blink rate and tone-related activation were measured on a per-interval basis and then related to each other, but that confuses me: if you want to get tone-related VTA activation for an interval, do you measure it in response to every tone in that interval, and then average the responses?

Doesn't the result in Figure 4e directly follow from the result shown in Figure 3e? If Figure 4e is only looking at post-tone responses (both blink count and VTA parameter estimates), then this relationship existing at the level of individual tones (as shown in Figure 3e) would necessarily imply that it would exist across sets of tones spanning an interval. Maybe the blink rate estimate for Figure 4e also includes periods outside of the 1.5s following each tone?

Sincerely,
Ken Norman (I sign all of my reviews)

(Remarks on code availability)

Reviewer #3

(Remarks to the Author)

This study tackles an important question—how dopaminergic activity shapes temporal distortions at event boundaries—using a clean auditory-boundary paradigm, fMRI, and eye-tracking. The behavioral task is elegant and the imaging/analysis pipeline is clearly documented, so the work has genuine potential impact. That said, several issues limit the strength of the current conclusions. First, the reported correlation between VTA activation and boundary-spanning temporal-distance ratings appears weak tendency; a direct statistical comparison of VTA-behaviour slopes for same- versus different-context pairs is not significant. Because VTA also tracks novelty and salience, the present data could reflect generic salience coding rather than segmentation-specific memory signals. Second, the dataset includes temporal-order-judgement trials, yet only distance ratings are analyzed. Showing whether VTA activity and blink rate relate (or fail to relate) to both memory probes is essential for clarifying what facet of “event memory” is being captured. Third, previous analyses of this same cohort highlighted locus-coeruleus and hippocampal contributions to event segmentation; integrating the current VTA results into that network—for instance by testing trial-wise covariation between VTA and hippocampal activity—would clarify whether dopamine merely registers contextual changes or actively modulates memory encoding. Fourth, because blink rate did not increase at event boundaries, it remains unclear whether the sharp VTA activation observed at those boundaries is directly tied to blink generation. Blinks might instead cluster before a boundary in anticipation, since the context length was fixed at eight items and participants could easily predict when the next boundary would occur. I recommend analyzing the blink-rate time course at a finer temporal resolution to test for changes immediately before and after boundaries. Addressing these points would substantially strengthen the manuscript and clarify whether dopaminergic processes genuinely drive boundary-related distortions in event memory.

(Remarks on code availability)

Version 1:

Reviewer comments:

Reviewer #1

(Remarks to the Author)

I believe that the authors did their best to address my comments. It looks like my feedback improved the scientific merit of the manuscript leading to new and interesting findings. My feedback also served to provide means for the authors to learn new methods and modify their approach. Thus I believe that the authors performed well scientifically and this manuscript deserves publication.

(Remarks on code availability)

Reviewer #2

(Remarks to the Author)

The authors have thoroughly addressed my prior set of concerns and I'm ready to sign off on the paper.

One very minor point: In the Supplementary Material, the authors say “Blinks in the earlier vs. later interval did not influence temporal memory predictions” – I would phrase this as “Blinks in the earlier vs. later interval did not differentially influence temporal memory predictions”

Another small note: In the rebuttal letter (in response to my comment #5) the authors say: "There were no significant differences between the relationships of LC and VTA activation at tone switches and DG pattern similarity for image pairs spanning those switches" ... but there were in fact differences, which are accurately described in the text, and also in the detailed response to this comment. So this is an issue with the text of the rebuttal but not an issue with the paper itself; there's no actionable item here.

Sincerely

Ken Norman (I sign all of my reviews)

(Remarks on code availability)

Reviewer #3

(Remarks to the Author)

The authors have carefully and thoroughly addressed all previous comments, providing additional analyses that strengthen the manuscript. The new results clarify that VTA activation and blink behavior relate specifically to the perceived time dilation between events, rather than to temporal order judgments or hippocampal representations. This provides novel evidence suggesting that dopaminergic processes play a key role in segmenting memories of distinct events.

(Remarks on code availability)

Response to Reviewers

We are grateful to the reviewers for their insightful comments and constructive suggestions. We believe that addressing their concerns and integrating their feedback has strengthened our manuscript significantly. Below, we provide point-by-point responses to each critique. Before detailing these responses, we highlight two general revisions:

- (1) We have revised our blink-related analyses to focus on **blink count** rather than blink rate. This change more accurately reflects blink behavior, particularly over short, seconds-long intervals. For example, a 3.5s interval containing 6 blinks could be artificially inflated to an estimate of 102.86 blinks/min, which isn't feasible. Using blink counts avoids this distortion and provides a more direct representation of the data. Importantly, all key results remain the same when using blink rate versus count, so this revision improves interpretability without altering our conclusions.
- (2) We have incorporated the requested follow-up analyses and additional details in an expanded **Supplementary Material** section. We chose to present these new figures and results separately from the main text to maintain focus on our core hypotheses and findings. This supplementary document helps provide a richer and more nuanced perspective on the data, clarifying the specificity of VTA effects and the temporal dynamics of blinking and memory organization.

Responses to Reviewer #1

Summary: *In this study authors explored the blink rate changes, VTA activity and short-term memory associative chunking (event boundaries and chunking intervals). While I found the study interesting, methods and approach valid, and the citations reliable, I have some concerns in experimental design and conclusions drawn from it. So I may not sure I would accept the paper for publication in its current state. Maybe the authors can help me to see the paper in a better light.*

Overall, I appreciate the work behind this study and that authors kindly provided open-access for their pipelines and data. If they can provide satisfactory answer to my comments and questions I can recommend their paper for publication.

I was able to see code under <https://osf.io/8y7hr/files/>.

Comment #1A: Major issue is primacy-recency effects: in short-term memory research, it is a well-known fact that the middle items in a sequence of item-lists are most likely to be forgotten. When a subject is asked a pair of objects which are 5-items apart among 8-item lists of chunks, the likelihood that this pair would be where one item falls on the middle section and the other one on the prior or recent section in the same chunk group, while it always falls as primacy-recency association between the separate chunks/events.

RESPONSE: We thank Reviewer #1 for raising this important point about potential confounds related to primacy and recency effects in list-learning paradigms. We believe these effects are unlikely to explain our memory findings due to several key aspects of the experimental design, detailed below:

1) Both item pair types were similarly susceptible to potential serial position effects. Our main comparison contrasted item pairs that spanned an event boundary (across ‘separate chunks’) with item pairs encountered within the same auditory context (in the same ‘chunk group’). Because item positions were crossed between the two pair types, both could be influenced by any ‘mini’ primacy or recency effects. This feature would effectively balance any confounding effects out. For example, as shown in the new **Supplementary Figure 1** below (top part), a boundary-spanning pair (yellow) could contain the final item in Event A (i.e., 8th item, likely to benefit from a recency effect) and a middle item in Event B (i.e., the 12th item; not likely to benefit from either primacy or recency). Thus, both boundary-spanning pairs and same-context pairs were susceptible to potential serial position influences, minimizing their impact on our key comparison. Additional details of this crossed pair structure are displayed in **Supplementary Figure 2 (the bottom part of figure below)**.

2) The task was designed to assess long-term memory rather than short-term memory. Our event sequence paradigm was specifically developed to probe long-term memory processes. During encoding, participants studied a continuous sequence of 32 items, which

could be *mentally* chunked into four 8-item sub-lists based on the auditory contexts. Even if participants treated these sub-events as mini-lists, it is unlikely that later item pairs in each chunk would benefit from a classic recency effect or working memory effects, as too much time would have passed between study and test. For example, the item in the 8th position in a list was not tested until after all 32 items were presented, well beyond the typical duration of short-term memory (~20s without rehearsal). Thus, such pairs would not plausibly benefit from a recency advantage.

Furthermore, a 45-s arrow distractor task was inserted between the end of encoding a list and the beginning of its corresponding temporal memory test. This delay and disruption task followed standard procedures in list-learning paradigms, serving to prevent list rehearsal and to reduce potential recency effects.

Investigating potential serial position effects on subjective distance memory. Previous list-learning studies have reported a ‘local mini-primacy’ effect, whereby item pairs presented at the beginning of each mini-event, or chunk, are remembered better for their objective temporal order (Pu et al., 2022). However, it remains unclear whether serial position-like effects extend to *subjective* temporal distance estimates. Prior work has primarily examined recognition or recall accuracy for individual items or their objective temporal order rather than their remembered distance.

To explore this possibility, we tested whether the serial position of tested item pairs during encoding predicted subsequent subjective temporal distance ratings. We began by focusing on same-context item pairs, which reflect memory for associations formed within a single auditory context, or event. There were two same-context item pairs tested from each 8-item event, allowing us to examine whether distance ratings differed according to the timing of those items (‘early’ versus ‘late’; paralleling analyses with temporal order memory in Pu et al., 2022).

As in the original analyses, we excluded the very first same-context pair, because it contained the very first item in the list and likely constituted a task-irrelevant event boundary. Using a cumulative link model with a categorical predictor for ‘early’ and ‘late’ position, we found that the relative timing of the same-context item pairs did not predict temporal distance ratings ($\beta = -.00049$, $SE = .041$, $z = -.012$, $p = .99$; see plot below). Therefore, there was no significant evidence of primary or recency ‘mini-list’ effects in temporal distance memory within contexts.

Next, we broadened our analysis to test whether item pair position across the *entire* list predicted temporal distance memory. Using another cumulative link model, we found a significant main effect of pair position ($\beta = .035$, $SE = .0081$, $z = 4.31$, $p < .001$), such that later pair positions in the lists were remembered as occurring farther apart in time than item pairs encountered early in the list (see **Supplementary Figure 5** below). Given the novelty and potential theoretical relevance of this finding to studies on temporal memory, we have included these results in the **Supplementary Material**. We have also added the following text to Results on **Page 5**:

"Interestingly, we also found a significant main effect of pair position ($\beta = .035$, $SE = .0081$, $z = 4.31$, $p < .001$), such that later pair positions in the lists were remembered as occurring farther apart in time than item pairs encountered early in the list (see **Supplementary Figure 5**). This pattern suggests a novel time-on-task-like effect for subjective distance ratings as the sequence unfolded."

Comment #1B: When people are asked to recall/recognise item-pairs, question was 'how far they are apart from each other'. It is possible that recognized items are more likely to be associated and perceived as closer to each other, while fanned-out items might be perceived as more distant.

RESPONSE: Investigating potential interactions between pair recognition and temporal estimates is indeed a very interesting question! However, in this experiment, we did not collect free recall or immediate recognition data, as these measures were not central to our primary goal of examining how event boundaries influence memory for time and event structure. We also anticipated that recognition performance would be very high due to the study-test block design: participants only needed to recognize 32 items after each list, likely resulting in near ceiling performance. Moreover, prior work suggests that encouraging strong item-focused encoding can impair temporal order memory by diverting attention away from the associations and connections between sequential information (e.g., Dubrow and Davachi, 2013). Thus, including such tests could have reduced overall temporal order accuracy in our experiment, which are reported elsewhere (Clewett et al., 2025).

Despite no immediate recognition test, there was a delayed item recognition test approximately 24 hours later. Thus, while somewhat indirect, we can loosely infer how strongly people encoded the individual items on Day 1. Analysis of this data revealed that item recognition

performance was quite good even a day later, suggesting that people likely had very strong item memory on Day 1 (M hit rate = .83; SD hit rate = .13; M corrected recognition = .71, SD corrected recognition = .16).

Following Reviewer #1's suggestion, we performed an exploratory analysis testing whether recognition of the item pairs was associated with temporal distance estimates for those same pairs. To do so, we conducted a recognition contingency analysis where item pairs were sorted according to whether both items were forgotten, only the first item was remembered, only the second item was remembered, or both items were remembered. Using a cumulative link model with contingency category modeled as a fixed-effect predictor, we found that there were no significant main effects of later recognition status on temporal memory ratings from the first day (all p s > .05; see plot below). These findings suggest item recognition status likely didn't influence retrospective estimates of temporal proximity, at least with recognition being inferred from a later timepoint.

Comment #2: I need more memory literature in the intro including Baddeley's theories.

RESPONSE: We thank Reviewer #1 for this helpful suggestion to incorporate additional literature on working memory. In response, we have added additional background on how dopaminergic processes contribute to working memory and how these processes align with models of event segmentation. However, we chose to limit discussion of Baddeley's theories because our sequence paradigm was specifically designed to examine long-term memory effects rather than short-term or working memory processes (see response to **Reviewer #1, Comment #1**). We have added the following text to the Introduction on **Page 3**:

"...and it is beneficial to update one's mental model of what is happening (Antony et al., 2021; Zacks et al., 2011). Indeed, this framework is consistent with dopamine's known role in working memory, which includes modulation of striatal activity to update ongoing representations (see D'Esposito & Postle, 2015) held in short-term memory storage (Baddeley & Hitch, 1974)."

"...dopamine may modulate this clock based on prediction errors (Mikhael & Gershman, 2019), which are theorized to trigger event segmentation

(Zacks et al., 2007) (see Fung, Sutlief, & Hussain Shuler, 2022). Many timing models also contain a working memory storage component that affects time perception (Fung, Sutlief, & Hussain Shuler, 2022) and is influenced by dopaminergic processes.”

Comment #3: Authors can benefit from P300 ERP literature related to contextual updating and its potential relationship with the VTA and ACC. I also encourage authors to report ACC region.

RESPONSE: We thank Reviewer #1 for their thoughtful suggestion to discuss potential links between the P300 and updating processes involving the VTA and anterior cingulate cortex (ACC). While a VTA-ACC pathway could indeed contribute to boundary processing, we chose to focus on local VTA activation to maintain a clear and streamlined narrative. Nevertheless, we agree that these other regions may be involved. To explore this possibility further, we conducted both an ROI-based analysis of the ACC and an exploratory whole-brain analysis to identify regions modulated by event boundaries.

The ACC is widely implicated in context updating and behavioral flexibility (e.g., Kolling et al., 2016). It has also been linked, at least in part, to generation of the P300 event-related potential (Crottaz-Herbette & Menon, 2006; see Sutton, Braren, & Zubin, 1965). Importantly, P300 activity has been linked to encountering novel event boundaries (Ezzyat & Clements, 2024), suggesting that the ACC may similarly signal contextual novelty in our event sequence paradigm. Motivated by this possibility, we tested whether the tone-switch boundaries elicited increased ACC activation during encoding. However, contrary to this prediction, Tone Type did not significantly predict tone-related ACC univariate (average) activation ($\beta = 2.26$, $SE = 1.53$, $t(9225.70) = 1.48$, $p = .14$).

The ACC is one among many key target brain areas that could support contextual updating. For example, other key nodes of the salience network, such as the anterior insula, are modulated by VTA activity and arousal (e.g., Seeley, 2019). Therefore, we also conducted an exploratory general linear model (GLM) analysis to examine whole-brain activation patterns modulated by event boundaries. Here, each tone was modeled as a 1-s stick regressor, each image was modeled as a 2.5-s stick regressor, and both were convolved with a double-gamma HRF. We also separated these regressors of interest into four event-related regressors: boundary tones, boundary images (first image that followed a tone switch), same-context tones, and same-context images (those in positions 2-8 in an event). The same physiological and movement noise regressors (e.g., motion, WM/CSF timeseries etc.) as our original LSS-GLM analyses were also included in this model. For this GLM, we included a 6-mm smoothing kernel to increase signal-to-noise ratio for activation patterns across the whole brain. The whole-brain contrast for boundary vs. same-context tones is displayed below:

The results of this GLM revealed many sensible patterns of event boundary-related brain activation, including increased activation in motor regions (M1, basal ganglia, etc.), goal-directed processing regions (lateral PFC, dorsal attention network), and auditory regions (auditory cortex, likely responding to the salience of the tone switch). Mirroring our ROI analysis, we did not observe any distinct clusters of activation in the ACC. While somewhat indirect, the lack of a boundary-related ACC effect aligns with the argument in the Discussion that prediction errors were not a critical mechanism of event segmentation in our paradigm. To maintain focus on our a priori predictions about neuromodulation, we only report these exploratory ACC ROI and whole brain results in this response letter.

Comment #4: Authors did not use false item-pairs in which items not memorized are shown, at all as a control condition. Nor did they use different distance metric (maybe in some cases 7-items distance) to test against the effect of recency.

RESPONSE: We thank Reviewer #1 for this opportunity to clarify important features of our experimental design. We chose not to use lures in the memory test for several reasons. First, the responses on the temporal distance memory test ('very close,' 'close,' 'far,' or 'very far') were inherently subjective. This was intentional, because it allowed us to examine distortions in temporal memory as opposed to objective accuracy. Second, given that participants studied only 32 items in each study-test block, participants would likely have close to (if not fully) ceiling performance on an immediate recognition test. Indeed, even after a 24-hr delay, we observed strong item recognition performance (mean hit rate = .83; *SD* hit rate = .13; corrected recognition = .71, *SD* corrected recognition = .16). For a short 45-s study-test delay, we would expect performance to be even higher. Additionally, including lure pairs would make recognition discrimination even easier, given that participants would only need to recognize that one of the items was new to make a correct rejection.

Regarding the objective spacing of the to-be-tested pairs, prior work demonstrates that event boundary effects on temporal distance memory are observed at varying intervals, including when to-be-tested item pairs contain two or four intervening items (Clewett et al., 2020). So, at least at these temporal gaps, segmentation effects appear to be insensitive to objective spacing during encoding. This points to the robustness of event segmentation processes in the Ezzyat-Dubrow-Davachi (EDD) paradigms (and in particular, the current auditory event sequence manipulation; Buonomano et al., 2023).

Limiting our comparisons to the same 5-item length was ideal because it enabled us to maximize the number of tested item pairs without any item repeats during the temporal memory test (see **Supplementary Figure 1**). This increased our condition bin sizes and maximized statistical power in our analyses. As described in our response to **Reviewer #1, Comment #1**, we also did not expect recency effects in the temporal distance results, given our memory test measures long-term memory, the crossed nature of the selected pairs across conditions, and the fact that we are querying subjective estimates of time and not objective accuracy.

Comment #5: It would be better to use longer time window to count blinks. I understand that they used 1.5s window because it was the shortest epoch, however I would use all the time segment after the beep till the first image of the next chunk was shown. They can split long and short windows and report separately.

RESPONSE: We thank Reviewer #1 for this helpful suggestion to separately analyze the different ISI lengths (1.5s, 2.5s, and 3.5s). To test this, we conducted separate linear mixed effects modeling analyses for the 1.5s ISIs, 2.5s ISIs, and 3.5s ISIs trials. We report the results here but not in the revised manuscript, given that the results largely replicate the original findings focused on the smallest shared epoch (1.5s) and were not pre-registered.

In brief, we replicated all our findings from the initial 1.5s of all ISIs to the full duration of each of the three ISI types (1.5s, 2.5s, and 3.5s). The one exception was for trial-level coupling between post-tone blink count and tone-related VTA activation. Specifically, we found that VTA activation predicted blink count during the 2.5s ISI type but not blink count during the 1.5s or 3.5s ISI types. We describe these analyses and statistical results in more detail below. Each model included random intercepts for participant ID.

Event boundaries predicted a momentary increase in blinking in all ISI types. For these models, Tone Type was modeled as a categorical fixed-effect predictor of blink count. We found that there was a main effect of Tone Type on local blink count in each of the ISI types: 1.5s ($\beta = .12$, $SE = .037$, $t(1988) = 3.26$, $p = .0011$), 2.5s ($\beta = .17$, $SE = .036$, $t(3190) = 4.78$, $p < .001$), and 3.5s ($\beta = .14$, $SE = .049$, $t(2083) = 2.95$, $p = .0032$). This replicated the finding that boundary tones were immediately followed by more blinks than same-context tones.

Tone-related VTA activation predicted local changes in blinking for the 2.5s ISI type. For these models, tone-related VTA activation (mean-centered) and Tone Type were modeled as continuous and categorical fixed-effect predictors of blink count, respectively. We found that there was a significant main effect of VTA activation on local blink count for the 2.5s ISI type ($\beta = .00057$, $SE = .00022$, $t(3116) = 2.67$, $p = .0077$), but not the 1.5s type ($\beta = .00022$, $SE = .00025$, $t(1941) = .86$, $p = .39$) or 3.5s type ($\beta = .000075$, $SE = .00030$, $t(2015) = .25$, $p = .80$). There were no significant VTA activation-by-tone-type interaction effects on blinking (all $ps > .05$). Thus, tone-related VTA activation predicted immediate blinking during intermediate-length ISIs (in addition to the first 1.5s of all ISI types, as previously shown in the original manuscript).

Local changes in blinking during all ISI types did not predict temporal distance memory. For these models, blink count (mean-centered) and Pair Type were modeled as continuous and

categorical fixed-effect predictors of distance memory, respectively. There were no significant main effects of local blink count on temporal distance ratings for any of the ISI types (all $ps > .05$). There were also no blink-by-pair type interaction effects on temporal distance ratings for any of the ISI windows (all $ps > .05$). This result mirrored our original finding that blinking during the first 1.5s of all ISIs did not predict later memory separation effects.

Comment #6: Authors mentioned LC. However, this area is more likely to be related to pupil size. If they are not going to report pupil size I would ask them to take it out of the paper. Or include pupil size.

RESPONSE: This is an excellent point! We briefly mentioned the LC in this manuscript because, alongside the VTA, the noradrenergic system has been theorized to support the tracking and updating of event representations (Zacks & Sargent, 2010). In the same dataset as the current study, we recently observed that tone-related LC activation at boundaries predicted poorer temporal order memory, an additional behavioral marker of memory separation (Clewett et al., 2025). Given that both the VTA and LC have been implicated in memory encoding and event segmentation, we included multiple LC-related measures, including pupil dilation, to test the specificity of the relationships between these brain regions and both memory and eye-tracking measures. Demonstrating such selectivity, such as a pupil-LC link but not pupil-VTA link, would strengthen our claims about the distinct contributions of these arousal-related neuromodulatory systems.

Because the focus of our paper was on the VTA and dopaminergic system, we have included these additional exploratory analyses in the **Supplementary Material**:

“Pupil dilation temporal principal component analysis (PCA). Prior work shows that context shifts elicit increased pupil dilation, suggesting that boundaries engage central arousal processes (Clewett et al., 2025). Pupil dilation, however, is complex and is mediated by multiple autonomic pathways and neuromodulatory systems (Reimer et al., 2016). Building on earlier work (Clewett et al., 2020), we use a temporal principal component analysis (PCA) to decompose boundary-related pupil dilations into its distinct temporal features, providing a unique opportunity to link event segmentation to different neural and behavioral effects. The temporal PCA on tone-evoked pupil dilations revealed three canonical pupil components identified in prior work, including a biphasic response that may index separate influences of parasympathetic and sympathetic nervous system regulation on pupil diameter (Clewett et al., 2020; Steinhauer and Hakerem, 1992). The temporal characteristics of these pupil components, including their latencies-to-peak and percent of explained variance, were as follows: (1) an early-peaking component (684 ms; 89.26% variance); (2) intermediate-peaking component (1,420 ms; 8.40% variance); and (3) slowly decreasing component (19.6 ms; 1.27% variance). For more details about these methods and results, see Clewett et al. (2025).

Testing the relationship between boundary-related VTA activation and three distinct temporal features of tone-evoked pupil dilation. In previously published work using this dataset, we found that pupil components #2 and #3 were both positively coupled with boundary-related LC activation (Clewett et al., 2025). Here, we tested whether the pupil components were also correlated with engagement of the VTA.

Using Spearman's rho correlations, we found that boundary-induced VTA activation was not significantly correlated with boundary-induced loading on pupil component #1 ($\rho = .14$, $p = .47$), pupil component #2 ($\rho = .11$, $p = .58$), or component #3 ($\rho = -.079$, $p = .69$; **Supplementary Figure 11**).

Comparing the relationships between boundary-related VTA vs. LC activation and pupil dilation. Next, we examined whether the coupling between brainstem activation and the three pupil components was specific to the LC using a Steiger's Z test. We found no significant differences in the linear relationships between LC and VTA activation and pupil component #1 ($z = -.44$, $p = .66$), pupil component #2 ($z = -.44$, $p = .66$), or pupil component #3 ($z = .48$, $p = .63$). Thus, while VTA activation did not relate to distinct temporal characteristics of pupil dilation at boundaries, we cannot conclude that neuromodulatory coupling with pupil responses was unique to LC activation."

Supplementary Figure 11. Boundary-related VTA activation was not associated with any of three temporal features of tone-evoked pupil dilations. Spearman's rho correlation plots showing that boundary-induced VTA activation was not significantly correlated with boundary-related engagement of pupil dilation component #1 (*left panel*), component #2 (*middle panel*), or component #3 (*right panel*). Individual dots represent each participants' data. X- and y-distributions also displayed. For more details about the temporal characteristics of these pupil components, see Figure 4 in Clewett et al. (2025).

Comment #7: There are a few outlier subjects who always pressed 'very far' or very close'. I do not think these subjects performed the task at all and should be taken out. You can also look at the box-test results and it is very likely that these subjects always pressed big or always small (ehre are the accuracy reports?) I understand that the sample size is already reduced but this will be more appropriate to do.

RESPONSE: We thank Reviewer #1 for bringing this to our attention, Upon closer inspection of the data, we did not identify any participants who exhibited this response pattern. Nonetheless, we appreciate the opportunity to provide readers with a more detailed audit of participants' behavior during encoding and retrieval. To summarize these findings, we have added two dot plots to the **Supplementary Material**.

Behavioral responses to object size judgment question during encoding. We first examined the distribution of participants' size judgment responses during encoding (i.e., responses to the orienting question "Is this object larger or smaller than a standard shoebox?"; see **Supplementary Figure 3** below). Participants were instructed to respond with either their left or right hand, depending on the side of the auditory tones (e.g., right ear = right hand). We did not observe any clear bias in these responses. Encoding accuracy is not reported because: 1) it was not central to the main analyses, and 2) accuracy could not be clearly defined, as some of the judgments were relatively subjective. For example, the size of a globe or banana could be accurately categorized as small or large, depending on each participant's perceptions of the objects. Additionally, our linear mixed effect modeling approach also helped account for atypical and variable responses. A key strength of this technique is that it controls for individual differences by including random intercepts for participant ID. Finally, the primary condition of interest (Pair Type: boundary-spanning vs. same-context pairs) was within-subject, so any participant-level differences in temporal distance ratings did not influence our findings. The distribution of participant-specific size judgments are displayed below:

Supplementary Figure 3. All participants used all four response options during the encoding orienting question. Response distributions for each participant to the simple orienting question: "Is this object larger or smaller than a standard shoebox?". For each object

image, participants were instructed to respond with a button box in their left or right hand, depending on the side of the preceding auditory tone (e.g., right ear = right hand).

Behavioral responses on the temporal distance memory test. Next, we examined participants' response distributions for the four options on the temporal distance memory test (see **Supplementary Figure 4** below). While two participants did not use one of the four response options (i.e., did not use 'very far'), none of the participants *only* used one of the response options. This response pattern suggests that all participants were likely engaged in the task. Without asking participants explicitly, we also cannot be certain whether their response patterns genuinely reflect their memory for temporal distance or whether they reflect a heuristic or lack of task engagement. Additionally, these potential trial exclusions were not pre-registered, so we believe it would be better practice to keep as much (sensible) data as possible.

Supplementary Figure 4. Participants generally used all four rating options on the temporal distance memory test. Response distributions for each participant on the temporal distance memory test. On each test trial, participants were presented with different pairs of objects from the prior sequence and asked to rate how far apart they thought each pair appeared in time ('very close', 'close', 'far', or 'very far'), despite their equivalent objective distance during encoding.

Comment #8: Demiral 2023 has interesting supplementary material; looking at a gambling task and explored button presses and blink timing. This might be helpful to notice how blinks wait for wrap-up of the motor action.

RESPONSE: We thank Reviewer #1 for directing our attention to these relevant findings. We have added the following text on **Page 13**:

“Context shifts also predicted brief increases in blinking, aligning with studies showing that blinks tend to occur near event transitions....Specifically, we found that these blinks occurred after boundary tones, not before them. This timing is consistent with evidence that blinking tends to be delayed when a motor response is required (Demiral et al., 2023), such as a button press response to an orienting question (“Is this object larger or smaller than a standard shoebox?”).”

Comment #9: Authors used multi-band imaging; I wonder whether collecting single band reference scan would improve preprocessing (authors did not mention they collected it).

RESPONSE: We appreciate Reviewer #1’s suggestion regarding the use of a single-band reference image in our fMRI analyses. Indeed, we did collect a single-band reference (which we’ll refer to as “SBRef”) image for each participant, but they were not explicitly used during preprocessing. The main benefit of an SBRef image is that it provides higher SNR, slight improvements in motion correction, and functional-anatomical co-registration. In our case, all our fMRI analyses were performed in participants’ native functional space, and motion correction was performed using the mean BOLD image for each run. We conducted a very rigorous motion correction procedure, which included MCFLIRT head motion parameters, fsl_motion_outlier volume censoring, and block-level exclusions for absolute framewise displacement more than 1mm. We also conducted boxplot outlier exclusions by-participant for noisy VTA activation datapoints, which would further filter spurious and noisy trials possibly influenced by motion.

Although we did transform an MNI-space ROI mask of the VTA into each participant’s native EPI space, this transformation relied on distortion-corrected EPI-to-anatomical registration procedures (i.e., BBR) and fieldmap correction, which provide accurate alignment even without an explicit SBRef (e.g., Glasser et al., 2013). Furthermore, the VTA mask was thresholded at 75% tissue-type probability to ensure high spatial specificity; this helps minimize any potential influence of small, residual misalignments during co-registration. Thus, we expect that the omission of the SBRef would have negligible effects on the spatial precision or statistical results. Given the substantial computational cost and time-intensive nature of reprocessing the entire dataset relative to the marginal benefits, we opted not to rerun the analyses (for clarification, this would include over 20,000 separate GLM’s: 10 blocks x 32 participants x 64 trials). We have clarified in the Methods that the SBRef images were collected but not used in preprocessing on **Page 18**:

“A single-band reference image was collected for each run of the task, but was not included during preprocessing.”

Response to Reviewer #2

Summary. *This paper explores the role of the dopaminergic (DA) system in driving time dilation effects that are commonly observed at event boundaries. To study these effects, the authors*

use a variant of the Ezzyat-DuBrow-Davachi event segmentation paradigm, where changes in the pitch of tones presented in between object pictures are used to trigger event boundaries. The paper reports several important findings: tone switches activate VTA; the magnitude of the VTA response at tone switches predicts the level of behavioral time dilation; tone switches predict increased blinks; VTA activation predicts blinking in general; and higher blink rates between to-be-judged items predict greater time dilation, but only for boundary-spanning pairs. Overall, I think this is an exciting package of results and I am very enthusiastic about the paper – but I do see some issues with the present version of the paper that I'd like the authors to address.

Broad critique: The first set of issues relates to how specific these results are to the DA system. Other results from this same study, focused on how the noradrenergic (NE) system is engaged at event boundaries, were written up in a separate manuscript that is presently in press at Neuron. This paper and the Neuron paper focus on almost completely distinct sets of dependent variables. The present paper focuses on VTA activation to tone switches, temporal distance judgments, and blink rate; the Neuron paper focused on LC activation to tone switches (indicative of the engagement of the NE system), temporal order judgments (“which of these two items came first”), pupil diameter, and hippocampal pattern similarity. The present paper, as noted above, found that VTA activation at boundaries relates to temporal distance judgments and blink rate. The Neuron paper found that LC activation at boundaries relates to temporal order judgments, pupil diameter, and hippocampal pattern similarity. Given these results, it is natural to wonder whether the DVs that relate to VTA activation also relate to LC activation, and vice-versa. The present paper addresses this in a brief section on page 11 where the authors show that LC activation is not significantly related to temporal distance ratings or blink rate, and that VTA activation is not related to temporal order judgments. While this is suggestive of a double dissociation between the two systems, there are some missing comparisons (outlined below), and also the authors do not report any exact statistics or effect sizes and they don't show any results plots comparing the effects. To more firmly establish this double dissociation, I suggest that the authors run the following analyses:

RESPONSE: We thank Reviewer #2 for the excellent suggestion to examine double dissociations between VTA and LC activation patterns and other key measures of interest (e.g., memory, pupil dilation, and blinking). Given the reviewer's comments on these topics are closely related, we address them point-by-point below. We have also added an overview of the motivation for these analyses and a short summary of the highlights of these analyses to Results on **Page 11-12**:

“Testing the specificity of blink and distance memory measures to the dopaminergic system. Recent findings from this same dataset showed that pupil-linked arousal and locus coeruleus (LC), a key hub of the brain's arousal and noradrenergic system, were associated with impairments in temporal memory (Clewett et al., 2025). An important open question is whether our current findings on subjective temporal memory and blinking are linked to arousal-related neuromodulation more generally. According to theoretical frameworks of event segmentation (Zacks & Sargent, 2010), both the dopaminergic and noradrenergic systems are well positioned to elicit a global updating signal when event transitions occur. However, it is possible that they contribute to memory separation (Rouhani et al., 2024) and blink behavior in different ways. We have also previously shown that LC activation at boundaries relates to more differentiated multivoxel activation patterns in left dentate gyrus

(DG), a hippocampal subregion that is important for memory encoding and in disambiguating overlapping neural representations (Clewett et al., 2025). Much work shows that dopaminergic modulation of hippocampal processing also plays an essential role in structuring and encoding episodic memories, making it another candidate input signal for memory separation at its target regions (Shohamy & Adcock, 2010).

To test the specificity of dopaminergic effects on temporal and hippocampal processing, we ran several mixed effects modeling analyses that complemented those in Clewett et al. (2025), which had focused only on LC-related effects. We then also performed follow-up modeling to test whether relationships between VTA activation, eye-tracking measures (pupil dilation and blinking), hippocampal pattern stability, and temporal memory differed from those of the LC (see **Supplementary Material** for details on methods and results). In brief, we found that LC activation was more strongly coupled with impairments in temporal order memory, an objective behavioral index of memory separation, than VTA activation. However, VTA activation was not significantly more coupled with distance ratings than LC activation, so we cannot make strong claims about specificity. On the other hand, blinking was more strongly coupled with VTA activation than LC activation, and more strongly coupled with distance memory than impairments in order memory.

When probing links to the stability of hippocampal multivoxel patterns across time, LC activation was significantly more coupled with pattern differentiation in left DG compared to VTA activation. This suggests that noradrenergic processes may have a stronger influence over ongoing event representations supported by the hippocampus. Finally, VTA activation was not significantly correlated with different features of boundary-evoked pupil dilation (see Clewett et al., 2025). For more details concerning all these analyses and results, see **Supplementary Material and Supplementary Figures 6-11.**”

Comment #1: It would be useful to show, statistically, that the relationship between VTA responses to tone switches and temporal distance is larger than the relationship between LC responses to tone switches and temporal distance (presently, the authors only show that the latter relationship is nonsignificant).

RESPONSE: In brief, we found no significant difference between the relationships of VTA and LC activation at tone switches with temporal distance memory. We have added the following text to **Supplementary Material**, including referring the reader to more detailed methods about the LC in Clewett et al. (2025):

“Locus coeruleus (LC) region-of-interest (ROI) definition. The anatomy of the LC was visualized in each participant using a fast spin echo (FSE) T1 MRI sequence, which is sensitive to neuromelanin and water content within LC neurons. Each participant’s left and right LC were hand-drawn separately by trained raters and then transformed into each participant’s run-level functional space to extract parameter estimates of trial-level LC activation. For more details about this MRI sequence, drawing procedures, and analysis methods, see Clewett et al. (2025).

Testing the relationship between tone-related LC activation and temporal distance ratings. To assess whether temporal distance memory could also be predicted by LC activation, we fit a cumulative link model where tone-related LC activation was mean-centered by participant and modeled as a fixed-effect predictor. Condition was modeled as a categorical variable, and we also included an interaction term between the two. Pair position at encoding was modeled as an integer fixed-effect predictor. Participant ID was modeled as a random effect. The results revealed no significant main effect of LC activation on temporal distance ratings ($\beta = .018$, $SE = .031$, $z = .59$, $p = .56$) nor significant LC-by-pair type interaction effect ($\beta = .018$, $SE = .031$, $z = .58$, $p = .56$) (see **Supplementary Figure 6** below for results for each pair type separately).

Comparing relationships between tone-related LC vs. VTA activation and distance memory. Next, we clarified whether the coupling between brainstem activation and temporal distance memory for boundary pairs was specific to the VTA. We performed a likelihood-ratio test that compared a full model with both VTA and LC activation as predictors with a restricted model that averaged VTA and LC activation as one predictor. The test revealed that the full model did not significantly improve fit over the restricted model ($\chi^2(1) = .83$, $p = .36$). Therefore, we cannot conclude that the relationship with temporal distance for boundary pairs is unique to VTA activation.”

Supplementary Figure 6. Tone-related LC activation did not predict subjective temporal distance memory between item pairs. (Left panel) Participant-level trendlines plotting the association between boundary-related LC parameter estimates (mean-centered) and temporal distance memory ratings (displayed as a continuous variable). (Right panel) Participant-level trendlines plotting the association between same-context LC parameter estimates and temporal distance memory ratings. Dark, bold lines represent the average linear trend across participants.

Comment #2: It would be useful to show, statistically, that the relationship between LC responses to tone switches and temporal order is larger than the relationship between VTA

responses to tone switches and temporal order (presently, the authors only show that the latter relationship is nonsignificant).

RESPONSE: We found that the relationship between LC activation at tone switches and temporal order memory was significantly stronger than that of order memory and VTA activation. This points to specificity in the effects of noradrenergic modulation on shaping objective impairments in order memory linked to event segmentation. We have added the following information to the **Supplementary Material**:

“Testing the relationship between tone-related VTA activation and temporal order memory. To determine whether temporal order accuracy could also be predicted by VTA activation, we fit a generalized linear mixed effects model that included tone-related VTA activation as a fixed-effect predictor of order accuracy (correct = 1, incorrect = 0). We found no significant main effect of VTA activation ($\beta = .037$, $SE = .035$, $z = 1.05$, $p = .30$) or VTA-by-pair type interaction effect ($\beta = .014$, $SE = .035$, $z = .40$, $p = .69$) on order accuracy (see **Supplementary Figure 7** below).

Comparing relationships between tone-related LC vs. VTA activation and order memory. Next, we clarified whether the coupling between brainstem activation and temporal order ratings for boundary pairs was specific to the LC. We performed a likelihood-ratio test that compared a full model with both VTA and LC activation as predictors with a restricted model that averaged LC and VTA activation as one predictor. To avoid a singular fit, these models were fit with a fixed effect representing the side of the screen containing the correct answer (i.e., left or right). The test revealed that the full model significantly improved fit over the restricted model ($\chi^2(1) = 9.21$, $p = .0024$). This finding suggests that the relationship between brain activation and temporal order memory for boundary-spanning pairs was specific to the LC.”

Supplementary Figure 7. Tone-related VTA activation did not predict temporal order memory accuracy. Participant-level trendlines plotting the null relationship between VTA parameter estimates (mean-centered) and order memory accuracy for boundary-spanning pairs (yellow; left) and same-context pairs (purple; right). For modeling, VTA parameter

estimates were mean-centered for each condition separately. Order memory accuracy is displayed as a continuous variable. Dark, bold lines represent the average linear trend across participants.

Comment #3: it would be useful to show, statistically, that the relationship between VTA responses to tones and blink rate is larger than the relationship between LC responses to tones and blink rate (presently, the authors only show that the latter relationship is nonsignificant).

RESPONSE: The relationship between VTA activation and extended blink count was significantly larger than that of LC activation. We have added the following text to the **Supplementary Material:**

“Testing the relationship between tone-related LC activation and blinking. To assess whether blinking behavior could also be predicted by LC activation, we fit linear mixed effects models. First, we examined local blink count. We found no significant main effect of LC activation on post-tone blink count ($\beta = -.026$, $SE = .018$, $t(7308.17) = -1.44$, $p = .15$; **Supplementary Figure 9, left panel**). There was also no significant LC-by-pair type interaction effect ($\beta = -.018$, $SE = .018$, $t(7308.32) = -.97$, $p = .33$).

Next, we examined temporally extended blink count between to-be-tested image pairs. We found no significant main effect of LC activation on extended blink count ($\beta = .00025$, $SE = .00025$, $t(2806) = 1.00$, $p = .32$; **Supplementary Figure 9, right panel**). There was also no significant LC-by-pair type interaction effect ($\beta = .00023$, $SE = .00025$, $t(2806) = .90$, $p = .37$).

Comparing relationships between tone-related VTA vs. LC activation and blinking. Next, we clarified whether the coupling between brainstem activation and blinking between to-be-tested pairs was specific to the VTA. We performed a linear hypothesis test that compared a full model with both VTA and LC activation as predictors with a restricted model in which the VTA and LC coefficients were equal. The test revealed that these coefficients were not equal ($\chi^2(1) = 6.95$, $p = .0084$), suggesting that the relationship between brain activation and temporally extended blink count was specific to the VTA.”

Supplementary Figure 9. Tone-related LC activation did not predict momentary or temporally extended changes in blink behavior. (*Left panel*) Participant-level trendlines plotting the null relationship between LC parameter estimates and post-tone blink count. (*Right panel*) Participant-level trendlines plotting the null relationship between LC parameter estimates and extended blink count. Dark, bold lines represent the average linear trend across participants. LC parameter estimates were mean-centered for each model separately.

Comment #4: It would be useful to show, statistically, that the relationship between LC responses to tone switches and pupil diameter is larger than the relationship between VTA responses to tone switches and pupil diameter.

RESPONSE: The results revealed no significant difference between boundary-induced LC-pupil and VTA-pupil coupling (see our response to **Reviewer #1, Comment #6**). We have added the following to text to the **Supplementary Material**:

“Pupil dilation temporal principal component analysis (PCA). Prior work shows that context shifts elicit increased pupil dilation, suggesting that boundaries engage central arousal processes (Clewett et al., 2025). Pupil dilation, however, is complex and is mediated by multiple autonomic pathways and neuromodulatory systems (Reimer et al., 2016). Building on earlier work (Clewett et al., 2020), we use a temporal principal component analysis (PCA) to decompose boundary-related pupil dilations into its distinct temporal features, providing a unique opportunity to link event segmentation to different neural and behavioral effects. The temporal PCA on tone-evoked pupil dilations revealed three canonical pupil components identified in prior work, including a biphasic response that may index separate influences of parasympathetic and sympathetic nervous system regulation on pupil diameter (Clewett et al., 2020; Steinhauer and Hakerem, 1992). The temporal characteristics of these pupil components, including their latencies-to-peak and percent of explained variance, were as follows: (1) an early-peaking component (684 ms; 89.26% variance); (2) intermediate-peaking component (1,420 ms; 8.40% variance); and (3) slowly decreasing component (19.6 ms; 1.27%

variance). For more details about these methods and results, see Clewett et al. (2025).

Testing the relationship between boundary-related VTA activation and three distinct temporal features of tone-evoked pupil dilation. In previously published work using this dataset, we found that pupil components #2 and #3 were both positively coupled with boundary-related LC activation (Clewett et al., 2025). Here, we tested whether the pupil components were also correlated with engagement of the VTA. Using Spearman's rho correlations, we found that boundary-induced VTA activation was not significantly correlated with boundary-induced loading on pupil component #1 ($\rho = .14$, $p = .47$), pupil component #2 ($\rho = .11$, $p = .58$), or component #3 ($\rho = -.079$, $p = .69$; **Supplementary Figure 11**).

Comparing the relationships between boundary-related VTA vs. LC activation and pupil dilation. Next, we examined whether the coupling between brainstem activation and the three pupil components was specific to the LC using a Steiger's Z test. We found no significant differences in the linear relationships between LC and VTA activation and pupil component #1 ($z = -.44$, $p = .66$), pupil component #2 ($z = -.44$, $p = .66$), or pupil component #3 ($z = .48$, $p = .63$). Thus, while VTA activation did not relate to distinct temporal characteristics of pupil dilation at boundaries, we cannot conclude that neuromodulatory coupling with pupil responses was unique to LC activation."

Supplementary Figure 11. Boundary-related VTA activation was not associated with any of three temporal features of tone-evoked pupil dilations. Spearman's rho correlation plots showing that boundary-induced VTA activation was not significantly correlated with boundary-related engagement of pupil dilation component #1 (*left panel*), component #2 (*middle panel*), or component #3 (*right panel*). Individual dots represent each participants' data. X- and y-distributions also displayed.

Comment #5: It would be useful to show, statistically, that the relationship between LC responses to tone switches and DG pattern similarity is larger than the relationship between VTA responses to tone switches and DG pattern similarity

RESPONSE: There were no significant differences between the relationships of LC and VTA activation at tone switches and DG pattern similarity for image pairs spanning those switches. We have added the following text to **Supplementary Material**, including referring the reader to the more detailed methods about hippocampal segmentation and pattern similarity analyses in Clewett et al. (2025):

“similarity fMRI analyses. Left and right hippocampal subfields CA2/3, dentate gyrus (DG), and CA1 were segmented from each participant’s high-resolution anatomical scan using Freesurfer 6.0 (<https://surfer.nmr.mgh.harvard.edu/>). Validated hippocampal ROIs were then co-registered to each participant’s native/run-specific functional space and thresholded at 0.2 to reduce spatial overlap between adjacent subfields.

For each of these hippocampal ROIs, we extracted activation patterns from the trial-unique beta maps produced by the LSS GLM, which modeled stimulus-specific activation patterns for all tones and images in a sequence. Here, we focused on the multivoxel patterns evoked by the image pairs as an index of hippocampal pattern stability across encoding, with more similar patterns reflecting representational stability and more dissimilar patterns reflecting temporal pattern separation. Hippocampal subfield pattern similarity scores were computed at the item pair level by correlating multivoxel patterns between each of the to-be-tested trial pairs from encoding. For more details, see Clewett et al. (2025).

Testing the relationship between boundary-related VTA activation and hippocampal pattern similarity across time. To determine whether VTA activation at boundaries predicted hippocampal pattern similarity for to-be-tested item pairs spanning those same boundaries, we fit a linear mixed effects model with six predictors of VTA activation at boundaries: left and right DG, CA1, and CA2/3 pattern similarity. Unlike the LC (Clewett et al., 2025), left DG pattern similarity did not significantly predict tone-induced VTA activation ($\beta = -6.70$, $SE = 51.44$, $t(1348.17) = -.13$, $p = .90$). There were also no significant main effects of right DG, left or right CA1, or left or right CA2/3 ($ps > .05$) (all subfield results are displayed in **Supplementary Figure 10**).

Comparing relationships between boundary-related LC vs. VTA activation and hippocampal pattern similarity. To directly compare the effect of left DG pattern similarity on boundary-related LC and VTA activation, we added brain region (LC vs. VTA) as an interaction term in the model. We observed a significant region-by-left DG interaction effect ($\beta = -414.35$, $SE = 120.49$, $t(2732) = -3.44$, $p < .001$). A simple slopes analysis revealed that the slopes of left DG pattern similarity significantly differed by brain region ($p < .001$), such that LC activation at boundaries was more strongly linked to left DG pattern similarity ($b = -841.9$) than VTA activation ($b = -13.2$). There were no other significant interaction effects ($ps > .05$).

In summary, these findings suggest that momentary increases in VTA activation at boundaries did not modulate pattern similarity in

hippocampal subfields. We also found that LC activation (vs. VTA) at boundaries was more strongly coupled with left DG pattern similarity for items spanning those boundaries, suggesting the LC may play a larger role in differentiating memory representations across time. Moreover, LC activation was selectively coupled with impairments in temporal order memory across boundaries, suggesting that the noradrenergic system might specifically modulate objective features of temporal memory.”

Supplementary Figure 10. Hippocampal pattern similarity for item pairs, including in the dentate gyrus, does not predict boundary-related VTA activation. Participant-level trendlines plotting the null relationships between hippocampal subfield pattern similarity for to-be-tested item pairs (in dentate gyrus, or DG, CA1, and CA2/3; left and right) and boundary-related VTA parameter estimates between those same item pairs. Dark, bold lines represent the average linear trend across participants. Each subfield’s pattern similarity values were mean centered separately and were all included as predictors of boundary-related VTA activation in the same linear mixed effects model.

Comment #6: In the methods section, the authors should either refer to the Neuron paper for methods relating to LC activation and pupil diameter measurements, or else they should copy those methods into the present paper

RESPONSE: We thank Reviewer #2 for these helpful suggestions. We have added the following text to Methods on **Page 20**:

“Alongside the blink estimates, we also computed pupil dilation responses to both boundary tones and same-context tones. A temporal principal component analysis (PCA) was applied to these tone-evoked pupil responses to dissociate distinct autonomic and functional components of pupil-related arousal. Details about pupil analyses and methods are reported elsewhere (Clewett et al., 2025), because the goal of this manuscript was to focus on the specificity of the relationship between brainstem nuclei activation and eye-tracking measures of neuromodulation.”

We have also added the following two passages to the **Supplementary Material**, where the relevant analyses are included:

“Locus coeruleus (LC) region-of-interest (ROI) definition. The anatomy of the LC was visualized in each participant using a fast spin echo (FSE) T1 MRI sequence, which is sensitive to neuromelanin and water content within LC neurons. Each participant’s left and right LC were hand-drawn separately by trained raters and then transformed into each participant’s run-level functional space to extract parameter estimates of trial-level LC activation. For more details about this MRI sequence, drawing procedures, and analysis methods, see Clewett et al. (2025).”

“Pupil dilation temporal principal component analysis (PCA). Prior work shows that context shifts elicit increased pupil dilation, suggesting that boundaries engage central arousal processes (Clewett et al., 2025). Pupil dilation, however, is complex and is mediated by multiple autonomic pathways and neuromodulatory systems (Reimer et al., 2016). Building on earlier work (Clewett et al., 2020), we use a temporal principal component analysis (PCA) to decompose boundary-related pupil dilations into its distinct temporal features, providing a unique opportunity to link event segmentation to different neural and behavioral effects. The temporal PCA on tone-evoked pupil dilations revealed three canonical pupil components identified in prior work, including a biphasic response that may index separate influences of parasympathetic and sympathetic nervous system regulation on pupil diameter (Clewett et al., 2020; Steinhauer and Hakerem, 1992). The temporal characteristics of these pupil components, including their latencies-to-peak and percent of explained variance, were as follows: (1) an early-peaking component (684 ms; 89.26% variance); (2) intermediate-peaking component (1,420 ms; 8.40% variance); and (3) slowly decreasing component (19.6 ms; 1.27% variance). For more details about these methods and results, see Clewett et al. (2025).”

Comment #7: The second set of issues relates to the blink rate results. The authors show that blink rate in the 1.5 sec post-tone-switch did not predict temporal distance ratings, and they also show that average blink rate between to-be-judged items predicts temporal distance ratings, but only when there’s an event boundary. Since blink rate within-1.5-sec-post-tone-switch is factored into the average blink rate between the to-be-judged items, this means that blinks in

some other time interval (i.e., not within 1.5 sec after the tone switch) are contributing to the observed relationship to time dilation. It seems like the authors could do more to nail down which parts of the interval between the to-be-judged items are driving the effect. For example, it seems unlikely that blinks in the part of the interval before the tone switch are driving the effect, and relatively more likely that blinks after the tone switch (i.e., between the tone switch and the appearance of the second to-be-judged item) are driving the effect – it would be straightforward to run this analysis. And if blinks after the tone switch are driving the effect, but blinks in the 1.5 sec immediately after the tone switch are not driving the effect, this suggests that blinks even later post-tone-switch are driving the effect. Are these blinks coming in response to the items? or in response to the later tones? In general, how often do participants blink outside of the 1.5 sec window following tones? Gaining a better understanding of which parts of the interval are driving the predictive relationship to time dilation would add to the overall story.

RESPONSE: We appreciate these very insightful suggestions! We agree it would be illuminating to dissect the data to identify the specific blink periods and stimuli that predict later distortions in temporal distance memory.

In brief, we found that blinking in the *later* portion of the window between to-be-tested pairs predicted subsequent temporal distance ratings. Specifically, blinking after tones during this later interval (including the tone switch) predicted greater time dilation between the to-be-tested pairs. We now summarize these findings on **Page 9-10**:

“Identifying which blink periods and stimuli predicted subsequent temporal distortions in memory. The results so far demonstrate that transient blink count in the 1.5 sec post-tone-switch did not predict temporal distance ratings between object pairs. Yet, the longer period of blinking between to-be-judged items did predict temporal distance ratings across an event boundary. Because the momentary boundary-related increase in blinking was encompassed within the larger temporal interval between pairs, it is likely the case that blinking in some other time part of the interval (i.e., not specifically within the 1.5s after the tone switch) was contributing to the observed relationship to time dilation. To explore this nuance of the data, we investigated which specific periods of extended blinking drove the time dilation effect. In brief, we found that the aggregation of tone-evoked blinking during the later portion of the inter-pair interval (i.e., from the tone switch to the appearance of the second to-be-judged item) predicted significant time dilation effects across boundaries ($\beta = .036$, $SE = .017$, $z = 2.11$, $p = .035$). For more details about these blink analyses and results, see the **Supplementary Material** and **Supplementary Figure 12.**”

In **Supplementary Material** (pasted below), we now describe these analyses and statistical results in more detail. To better capture the richness and temporal dynamics of blink behavior, we also provide summary statistics of blinking across time.

“Identifying which blinking periods and types of stimulus-elicited blinked predicted time dilation in memory. In this set of analyses, we aimed to identify the specific blink periods and stimuli that predict later distortions in temporal distance memory.

Average patterns of blinking across the inter-pair windows. To capture temporally dynamic patterns of blinking across the event

sequence, we first divided each inter-item window into two coarse-grained intervals: 1) the interval *before* the reference tone (M duration = 12.28s); and 2) the interval *after* the reference tone (including that tone; M duration = 20.87s). For more details, see **Supplementary Figure 12** below. On average, participants blinked 8.83 times before the reference tone (M boundary pairs = 8.78 blinks; M same-context pairs = 8.87 blinks) and 15.43 times after the reference tone (M boundary pairs = 15.77 blinks; M same-context pairs = 15.10 blinks).

Next, given that the interval *after* the reference tone carried key contextual information in this task (e.g., that the auditory-task context had or had not changed), we divided this interval further. Specifically, we identified even more fine-grained intervals associated with each stimulus: 1) images; and 2) tones (**Supplementary Figure 12**). We observed that most blinks occurred after images (M overall = 12.56 blinks across images; M boundary pairs = 12.77 blinks; M = same-context pairs = 12.34 blinks) compared to tones (M overall = 4.65 blinks across tones; M boundary pairs = 4.76 blinks; M = same-context pairs = 4.54 blinks). This is a sensible result, given that images have a longer duration and subsequent ISI compared to tones.

Relating specific blink periods to temporal distance memory. Here, we conducted linear mixed-effects models to test which stimulus-evoked blinks and periods of blinking were significantly predictive of temporal distance ratings.

Blinks in the earlier vs. later interval did not influence temporal memory predictions. First, we tested whether the interval before versus after the reference tone drove this effect. There was no significant interaction between blink count, Interval Type (before vs. after reference tone), and Pair Type (boundary vs. same-context) on distance memory ($\beta = .0015$, $SE = .0032$, $z = .45$, $p = .65$).

Focusing on the later interval, blinking after a tone switch vs. no switch predicted greater time dilation in memory. While the prior interaction effects were null, we next focused specifically on the blink window *after* the reference tone, because this carried the critical information about the auditory-task context. In this period, we found a significant interaction between blink count and Pair Type on distance memory ($\beta = .0085$, $SE = .0040$, $z = 2.12$, $p = .034$). To break down this interaction effect, we then examined the two Pair Types separately. For boundary pairs, there was a marginally significant main effect of post-tone switch blink count on distance memory ($\beta = .011$, $SE = .0060$, $z = 1.87$, $p = .061$), such that more blinking after the tone switch predicted greater subsequent time dilation between the to-be-tested object pairs. In contrast, for same-context pairs, there was no significant effect of post-reference tone blink count on distance memory ($\beta = -.0057$, $SE = .0055$, $z = -1.04$, $p = .30$). Therefore, blinks after the reference tone predicted later time dilation only when there had been a tone switch, denoting a new auditory context.

Investigating whether image- or tone-evoked blink effects in the later inter-pair interval predicted time dilation in memory. Zooming in on this later interval further, we asked whether stimulus type mattered; that is, whether blinks following either *images* or *tones* were the driving factor behind changes in temporal distance ratings. We found that there were no interaction effects between blink count, Stimulus Type (Images vs. Tones), and Pair Type on distance memory (all p s > .05). Thus, at least during the critical latter part of the inter-pair interval, blinks following images vs. tones did not relate to temporal distance memory.

Given the lack of an interaction effect and that tones carried the important contextual information during the task, we next focused on tone-related blinking alone in the later inter-pair interval. We found a significant interaction between Pair Type and total blink count following tones on distance memory ($\beta = .031$, $SE = .012$, $z = 2.63$, $p = 0.0086$). To break down this interaction effect, we analyzed the two Pair Types separately. For boundary pairs, there was a significant main effect of total blink count following tones on distance memory ($\beta = .036$, $SE = .017$, $z = 2.11$, $p = .035$), such that more blinking after the tone switch predicted greater time dilation between the to-be-tested object pairs. In contrast, for same-context pairs, there was no significant effect of total blink count following tones on distance memory ($\beta = -.025$, $SE = .016$, $z = -1.56$, $p = 0.12$). Together, these findings suggest that the coupling between blinking and time dilation in memory was driven by tone switches, the stimulus that carried the critical signal denoting an event boundary.

In summary, our analyses showed that blinks following a tone switch (versus no switch) predicted greater time dilation in memory between the to-be-tested object pairs. This link between blinking and memory distortion was related to the occurrence of tones, suggesting that event boundaries play an important role in triggering dopaminergic processes that shape later memory separation effects.”

Supplementary Figure 12. Dividing the window between to-be-tested pairs into specific intervals of interest. Example intervening window for each to-be-tested image pair (see yellow arrow connecting pair of squares). Each window was approximately 32.5s long, containing 5

images and 4 tones total. These windows can be divided into coarse- and fine-grained intervals of interest. The coarse-grained intervals are separated by the reference tone (red dashed line), which is the tone of interest between each to-be-tested pair. For boundary-spanning pairs (example shown here), this was the tone switch that denoted an event boundary. For same-context pairs, this was a position-matched tone (for more details, see **Supplementary Figure 1, bottom panel**). Therefore, the two resulting intervals are as follows: (1) *Before* the reference tone (blue), from the onset of the first to-be-tested image to the onset of the reference tone; and (2) *After* the reference tone (orange), from the onset of the reference tone to the offset of the second to-be-tested image. Additionally, the window can be segmented further into fine-grained intervals that are associated with each individual stimulus: (1) Image intervals (brown), from the onset of the image to the end of the subsequent ISI (2.5s + variable + 0.5s); and (2) Tone intervals (gray), from the onset of the tone to the end of the subsequent ISI (variable). The final image interval (brown) extends partially outside of the “*After* the reference tone” (orange) interval, as this image interval also contains the ISI after the final image offset.

MINOR ISSUES

Comment #8: p. 12: “elevated blink rates at event boundaries predicted greater time dilation”... this is not an accurate description of the finding

RESPONSE: We thank Reviewer #2 for pointing this out. We have edited the passage on **Page 12** to be more accurate. We now also refer to blink *counts* rather than blink rates to avoid making large inferences and transformations to the blink data, as described in the general comment at the top of this letter.

“...elevated blink counts across event boundaries predicted greater time dilation between item pairs spanning those same boundaries.”

Comment #9: p. 12: “VTA responses to boundary tones... predicted transient increases in blinking” – this is a bit misleading, as VTA responses to non-boundary tones also predicted transient increases in blinking (fig. 3e)

RESPONSE: Thank you for catching this! We have updated this description on **Page 13:**

“Moreover, VTA responses to tones generally predicted transient increases in blinking.”

Comment #10: For analyses relating average blink rate between the to-be-judged items and tone-related VTA activation, when was tone-related VTA activation measured (Figure 4e)? My understanding was that both blink rate and tone-related activation were measured on a per-interval basis and then related to each other, but that confuses me: if you want to get tone-related VTA activation for an interval, do you measure it in response to every tone in that interval, and then average the responses?

RESPONSE: This is an excellent question and provides a valuable opportunity to clarify the structure of our linear mixed-effects modeling analyses. The main goal of these analyses was to

examine how trial-level variations in tone-evoked VTA activation and blinking behavior related to corresponding changes in temporal distance memory for item pairs spanning those same moments. To illustrate this logic, we have added **Supplementary Figure 1** (pasted below), which clarifies the task structure and the setup of these trial-level analyses.

Capturing VTA activation at a single tone between to-be-tested image pairs. The most informative way to test these neurophysiological-memory associations was to measure VTA activation specifically during the boundary (switch) tones. We focused on this period because it captures the key moment of event updating/segmentation, rather than the broader activity across the entire inter-pair interval. Including all tone-related VTA responses within the full boundary-spanning pair window could obscure these transient boundary effects, thereby reducing sensitivity to the VTA-memory relationship of interest.

To test this specific prediction, we correlated boundary-trial brain responses to distance memory for their specific to-be-tested item pairs using cumulative link models. For example, if we extracted VTA activation for the tone switch following the 8th item in a list (the first event boundary in a sequence), we were interested in whether this response predicted impaired order memory for its corresponding boundary-spanning item pairs (the pairs in positions 6 → 10 and 8 → 12 in the list; see **top panel of Supplementary Figure 1** below). For same-context pairs, we then position-match these comparisons by selecting tones from those same positions within the pairs (see **bottom panel of Supplementary Figure 1** below). The figure below illustrates the specific positions of the to-be-tested pairs in the list and the location of the reference tones where we measured/examined tone-evoked VTA activation.

Investigating mean tone-related VTA activation between to-be-tested image pairs. Even though we were targeting local effects of boundaries, it is certainly possible that the aggregation of VTA signals within the entire boundary-spanning window could drive memory distortions. To test this, we extracted and modeled the mean VTA activation across all tones in a pair interval as predictors of temporally extended blink behavior across those same windows. In brief, we found that averaged VTA activation over several tones does not carry the same predictive power as local VTA activation. Below, we describe these analyses and statistical results in more detail.

Mean tone-related VTA activation in the earlier vs. later interval did not influence blink behavior. First, we tested whether mean VTA activation for all tones *before* versus *after* the reference tone predicted extended blink count across the entire inter-pair window. A linear mixed effects model revealed no significant interaction between mean VTA activation, Interval Type (Before or After reference tone), and Pair Type (Boundary vs. Same-Context) on temporally extended blink count ($\beta = -.00045$, $SE = .0015$, $t(4005) = -.30$, $p = .76$).

After the reference tone. However, given that the interval *after* the reference tone carried the critical contextual updating signal (e.g., that the context had or had not changed), we focused on this latter part of the interval. Specifically, we tested whether mean VTA activation for all tones encountered *after* the reference tone (*including* that reference tone itself) predicted extended blink count across the entire window. There was no significant main effect of mean tone-related VTA activation ($\beta = .0011$, $SE = .0018$, $t(2791) = .59$, $p = .55$) or VTA-pair type interaction effect ($\beta = .0014$, $SE = .0018$, $t(2791) = .79$, $p = .43$) during this later window on extended blink count.

Before the reference tone. Next, we returned to test whether mean VTA activation for all tones encountered *before* the reference tone predicted extended blink count across the entire window. As before, there was no significant main effect of mean tone-related VTA activation (β

= .0025, $SE = .0023$, $t(1187) = 1.07$, $p = .28$) or VTA-pair type interaction effect ($\beta = .0025$, $SE = .0024$, $t(1188) = 1.06$, $p = .29$) on extended blink count.

Mean tone-related VTA activation across the entire window did not predict blink behavior. Finally, we tested whether mean VTA activation across the entire window (i.e., agnostic to the reference tone) predicted extended blink count for that window. Similarly, there was no main effect of mean tone-related VTA activation ($\beta = .0013$, $SE = .0022$, $t(2821) = .62$, $p = .54$) or VTA-by-pair type interaction effect ($\beta = .0024$, $SE = .0022$, $t(2821) = 1.12$, $p = .26$) on extended blink count.

In summary, it appears that momentary increases in VTA activation – that is, at a single, behaviorally relevant tone – was coupled with prolonged blinking patterns, while the average of repeated tone-induced VTA responses was not. This finding points to the importance of discrete boundary-related engagement of the dopaminergic system on blink behavior and memory.

Supplementary Figure 1. Item pair structure across the event sequence task. (*Top panel*) Each sequence contained 32 images of everyday objects. A pure tone was played either in participants' left ear or right ear for 8 successive items. It then switched to the other ear and changed in pitch. The side/tone then repeated for another 8 items before

switching ears again and so on. Colored squares denote the item pair positions that were subsequently queried during the temporal memory tests after each list. The arrow connections indicate which item pairs were later tested together, in the memory test. There were always three intervening items between each to-be-tested pair during encoding. Purple squares show the pair positions for same-context trials, or to-be-tested item pairs that were presented with the same tone. Yellow squares show the pair positions for boundary trials, or to-be-tested item pairs that spanned an intervening tone switch. Vertical dashed dark yellow lines indicate the positions of “event boundaries”, or the three auditory tone switches in each list. For the linear mixed effects modeling and cumulative link modeling analyses, we aimed to relate trial-level estimates of tone-evoked VTA activation to both temporal distance memory ratings and blinking behavior associated with their corresponding item pairs. To align these two measures appropriately, we specifically focused on VTA parameter estimates (derived from BOLD signal model fits in the GLM) evoked by event boundaries and same-context tones position-matched to those locations (all vertical gray lines). (*Bottom panel*) Example of position-matching between tone-evoked VTA activation between conditions. This illustration shows how the specific tones analyzed for VTA activation were position-matched based on the location of event boundaries. This provided a critical reference point for controlling timing-related effects of VTA activation relative to the positioning of the to-be-tested item pairs spanning those moments. In this example section from an item sequence, the first boundary tone occurred after the 8th item in the list. This means that the boundary occurred after two items for the item from the first pair that spanned those boundaries (i.e., item in position 6), and immediately after the first item in the second boundary-spanning item pair (i.e., item in position 8). To position-match these timing effects in the same-context condition, the two same-context pairs were also temporally aligned with the same sampling points. That is, the relative positioning of the selected tones in the analysis was the same as the two boundary-spanning pairs for each event boundary in the list.

Comment #11: Doesn't the result in Figure 4e directly follow from the result shown in Figure 3e? If Figure 4e is only looking at post-tone responses (both blink count and VTA parameter estimates), then this relationship existing at the level of individual tones (as shown in Figure 3e) would necessarily imply that it would exist across sets of tones spanning an interval. Maybe the blink rate estimate for Figure 4e also includes periods outside of the 1.5s following each tone?

RESPONSE: We thank Reviewer #2 for the opportunity to clarify this aspect of our analyses. To help clarify this point, we highlight **Supplementary Figure 2** (pasted below), which provides a helpful schematic for analyses such as those described in Figure 4e. Here, we found that a momentary increase in VTA activation (i.e., at a single tone) predicted more blinking across a temporally extended window. This window, approximately 32.5s long, contains images, tones, and intervening ISIs.

In contrast, the analysis from Figure 3e found that a momentary increase in VTA activation (i.e., at a single tone) predicted more blinks directly following that tone (i.e., 1.5s). Therefore, a

relationship existing at the level of individual tones does not necessarily imply that it would exist across the entire window.

Supplementary Figure 2. Detailed schematic of position labeling and matching between conditions. Several analyses examined how indirect measures of brief dopaminergic processes during encoding (i.e., tone-related VTA activation or post-tone blink count) predicted larger-scale outcomes (i.e., memory for temporal distance or blink rate between to-be-tested pairs). For these analyses, one intervening tone was selected for each to-be-tested pair in the event encoding sequence. Arrow connections represent to-be-tested pairs, including same-context pairs (purple) and boundary-spanning pairs (yellow). For boundary-spanning pairs, the selected tone was the tone switch that denoted an event boundary (red circle). Depending on the particular boundary-spanning pair, the to-be-tested items fell either 3 positions before (-3) and 2 positions after (+2) the tone switch or 1 position before (-1) and 4 positions after (+4) the tone switch. For same-context pairs, the selected tone (purple circle) matched these positions (i.e., -3 and +2 or -1 and +4).

Response to Reviewer #3

Summary. This study tackles an important question—how dopaminergic activity shapes temporal distortions at event boundaries—using a clean auditory-boundary paradigm, fMRI, and eye-tracking. The behavioral task is elegant and the imaging/analysis pipeline is clearly documented, so the work has genuine potential impact. That said, several issues limit the strength of the current conclusions.

Comment #1: First, the reported correlation between VTA activation and boundary-spanning temporal-distance ratings appears weak tendency; a direct statistical comparison of VTA–behaviour slopes for same- versus different-context pairs is not significant. Because VTA also tracks novelty and salience, the present data could reflect generic salience coding rather than segmentation-specific memory signals.

RESPONSE: We thank Reviewer #3 for raising this thoughtful alternative interpretation regarding salience-coding mechanisms. Indeed, because we did not observe a significant interaction effect between VTA activation and Pair Type on temporal distance ratings, we cannot conclude that the VTA-memory relationship is unique to boundary-spanning pairs. However, it is important to note that the significant VTA-memory coupling for boundary pairs emerged at the trial level, because we were performing linear mixed effects modeling. This finding suggests that boundary-evoked VTA activation was meaningfully related to memory encoding/distortion rather than salience coding more broadly; namely, the likelihood of time dilation occurring between a boundary-spanning pair was higher *when* the VTA responded more strongly to its intervening boundary tone.

It is also worth noting that tests for pair-type interactions were somewhat constrained by the task design. Specifically, linking tone-related VTA activation to distance memory for a to-be-tested image pair required selecting an intervening ‘reference tone’ within each inter-pair interval. For boundary-spanning pairs, this reference was clearly defined: it was the tone switch signaling the event boundary. For same-context pairs, the reference was the position-matched tone (for details, see **Supplementary Figure 2**, pasted below). Nevertheless, it is possible that other timepoints within the same-context intervals might reveal different relationships between tone-evoked VTA activation and distance memory, both within the same-context condition and across pair types. To address this possibility and clarify interpretive limits, we have added the following text to the Discussion on **Page 13-14**:

“It is possible that this VTA-memory relationship reflects processes beyond boundary-specific signaling, such as general salience or updating mechanisms. Nevertheless, because the coupling between boundary-evoked VTA activation and temporal memory emerged at the trial level, our results suggest that transient dopaminergic responses at event boundaries might meaningfully shape how temporal distance is encoded. We also note that interpreting VTA-memory relationships for same-context pairs is inherently constrained by choosing a single tone to index VTA activation: the reference tone that was position-matched to the tone switch between boundary-spanning pairs. It remains possible that there are activity patterns at other timepoints within this same-context pair interval that could yield a different pattern of results.”

Supplementary Figure 2. Detailed schematic of position labeling and matching between conditions. Several analyses examined how indirect measures of brief dopaminergic processes during encoding (i.e., tone-related VTA activation or post-tone blink count) predicted larger-scale outcomes (i.e., memory for temporal distance or extended blink count between to-be-tested pairs). For these analyses, one intervening tone was selected for each to-be-tested pair in the event encoding sequence. Arrow connections represent to-be-tested pairs, including same-context pairs (purple) and boundary-spanning pairs (yellow). For boundary-spanning pairs, the selected tone was the tone switch that denoted an event boundary (red circle). Depending on the particular boundary-spanning pair, the to-be-tested items fell either 3 positions before (-3) and 2 positions after (+2) the tone switch or 1 position before (-1) and 4 positions after (+4) the tone switch. For same-context pairs, the selected tone (purple circle) matched these positions (i.e., -3 and +2 or -1 and +4).

Comment #2: Second, the dataset includes temporal-order-judgement trials, yet only distance ratings are analyzed. Showing whether VTA activity and blink rate relate (or fail to relate) to both memory probes is essential for clarifying what facet of “event memory” is being captured.

RESPONSE: We thank Reviewer #3 for raising this excellent point about narrowing in on the specificity of VTA/blink and temporal distance effects. We have now conducted these follow-up analyses and added the following to **Supplementary Material**:

For VTA activation vs. temporal order memory:

“Testing the relationship between tone-related VTA activation and temporal order memory. To determine whether temporal order accuracy could also be predicted by VTA activation, we fit a generalized linear mixed effects model that included tone-related VTA activation as a fixed-effect predictor of order accuracy (correct = 1, incorrect = 0). We found no significant main effect of VTA activation ($\beta = .037$, $SE = .035$, $z = 1.05$, $p = .30$) or VTA-by-pair type interaction effect ($\beta = .014$, $SE = .035$, $z = .40$, $p = .69$) on order accuracy (see **Supplementary Figure 7** below).”

“Comparing relationships between temporal order vs. distance memory and VTA activation. To compare the magnitude of the

relationship between tone-related VTA activation and temporal memory for boundary pairs, we compared the standardized coefficients for VTA activation between the distance and order memory models. To avoid a singular fit, the order memory model was fit with a fixed effect representing the side of the screen containing the correct answer (i.e., left or right). The standardized coefficient for VTA activation in the distance memory model ($\beta^* = .10$, 95% CI: [.01, .20]) was not significantly different than in the order memory model ($\beta^* = .05$, 95% CI: [-.05, .16]). Thus, while VTA activation did not relate to temporal order memory, we cannot conclude that VTA coupling with memory was unique to temporal distance memory.”

Supplementary Figure 7. Tone-related VTA activation did not predict temporal order memory accuracy. Participant-level trendlines plotting the null relationship between VTA parameter estimates (mean-centered) and order memory accuracy for boundary-spanning pairs (yellow; left) and same-context pairs (purple; right). For modeling, VTA parameter estimates were mean-centered for each condition separately. Order memory accuracy is displayed as a continuous variable. Dark, bold lines represent the average linear trend across participants.

For extended blink count vs. temporal order memory:

“Testing the relationship between blinking and temporal order memory. To determine whether temporal order accuracy could also be predicted by blinking, we fit a generalized linear mixed effects model that included extended blink count as a fixed-effect predictor of order accuracy (correct = 1, incorrect = 0). We found no significant main effect of extended blink count ($\beta = -.0024$, $SE = .0044$, $z = -.55$, $p = .58$) or blink count-by-pair type interaction effect ($\beta = -.0052$, $SE = .0044$, $z = -1.18$, $p = .24$) on order accuracy (see **Supplementary Figure 8** below).

Comparing relationships between temporal order vs. distance memory and blinking. To compare the magnitude of the relationship between blinking and temporal memory ratings for boundary pairs, we

compared the standardized coefficients for blinking between the distance and order memory models. To avoid a singular fit, the order memory model was fit with a fixed effect representing the side of the screen containing the correct answer (i.e., left or right). The standardized coefficient for blinking in the distance memory model ($\beta^* = .16$, 95% CI: [.06, .27]) was significantly greater than in the order memory model ($\beta^* = -.07$, 95% CI: [-.18, .04]). This finding suggests that the relationship between blinking and memory for boundary-spanning pairs was greater for temporal distance memory than order memory.”

Supplementary Figure 8. Sustained blinking between to-be-tested item pairs did not predict memory accuracy for the temporal order of those items. Participant-level trendlines plotting the null relationship between extended blink count (mean-centered) and order memory accuracy for boundary-spanning pairs (yellow; left) and same-context pairs (purple; right). For modeling, blink counts were mean-centered for each condition separately. Order memory accuracy is displayed as a continuous variable. Dark, bold lines represent the average linear trend across participants.

Comment #3: Third, previous analyses of this same cohort highlighted locus-coeruleus and hippocampal contributions to event segmentation; integrating the current VTA results into that network—for instance by testing trial-wise covariation between VTA and hippocampal activity—would clarify whether dopamine merely registers contextual changes or actively modulates memory encoding.

RESPONSE: We thank Reviewer #3 for this helpful analysis suggestion, as it would strengthen our interpretations about VTA modulation of hippocampal function networks.

To address this possibility (see our response to **Reviewer #2, Comment #5**), we have added the following to **Supplementary Material**, referring readers to the more complex hippocampal segmentation and pattern similarity method details in Clewett et al. (2025):

Hippocampal pattern similarity fMRI analyses. Left and right hippocampal subfields CA2/3, dentate gyrus (DG), and CA1 were segmented from each participant's high-resolution anatomical scan using Freesurfer 6.0 (<https://surfer.nmr.mgh.harvard.edu/>). Validated hippocampal ROIs were then co-registered to each participant's native/run-specific functional space and thresholded at 0.2 to reduce spatial overlap between adjacent subfields.

For each of these hippocampal ROIs, we extracted activation patterns from the trial-unique beta maps produced by the LSS GLM, which modeled stimulus-specific activation patterns for all tones and images in a sequence. Here, we focused on the multivoxel patterns evoked by the image pairs as an index of hippocampal pattern stability across encoding, with more similar patterns reflecting representational stability and more dissimilar patterns reflecting temporal pattern separation. Hippocampal subfield pattern similarity scores were computed at the item pair level by correlating multivoxel patterns between each of the to-be-tested trial pairs from encoding. For more details, see Clewett et al. (2025).

Testing the relationship between boundary-related VTA activation and hippocampal pattern similarity across time. To determine whether VTA activation at boundaries predicted hippocampal pattern similarity for to-be-tested item pairs spanning those same boundaries, we fit a linear mixed effects model with six predictors of VTA activation at boundaries: left and right DG, CA1, and CA2/3 pattern similarity. Unlike the LC (Clewett et al., 2025), left DG pattern similarity did not significantly predict tone-induced VTA activation ($\beta = -6.70$, $SE = 51.44$, $t(1348.17) = -.13$, $p = .90$). There were also no significant main effects of right DG, left or right CA1, or left or right CA2/3 ($ps > .05$) (all subfield results are displayed in **Supplementary Figure 10**).

Comparing relationships between boundary-related LC vs. VTA activation and hippocampal pattern similarity. To directly compare the effect of left DG pattern similarity on boundary-related LC and VTA activation, we added brain region (LC vs. VTA) as an interaction term in the model. We observed a significant region-by-left DG interaction effect ($\beta = -414.35$, $SE = 120.49$, $t(2732) = -3.44$, $p < .001$). A simple slopes analysis revealed that the slopes of left DG pattern similarity significantly differed by brain region ($p < .001$), such that LC activation at boundaries was more strongly linked to left DG pattern similarity ($b = -841.9$) than VTA activation ($b = -13.2$). There were no other significant interaction effects ($ps > .05$).

In summary, these findings suggest that momentary increases in VTA activation at boundaries did not modulate pattern similarity in hippocampal subfields. We also found that LC activation (vs. VTA) at boundaries was more strongly coupled with left DG pattern similarity for items spanning those boundaries, suggesting the LC may play a larger role in differentiating memory representations across time. Moreover, LC activation was selectively coupled with impairments in temporal order

memory across boundaries, suggesting that the noradrenergic system might specifically modulate objective features of temporal memory.”

Supplementary Figure 10. Hippocampal pattern similarity for item pairs, including in the dentate gyrus, does not predict boundary-related VTA activation. Participant-level trendlines plotting the null relationships between hippocampal subfield pattern similarity for to-be-tested item pairs (in dentate gyrus, or DG, CA1, and CA2/3; left and right) and boundary-related VTA parameter estimates between those same item pairs. Dark, bold lines represent the average linear trend across participants. Each subfield’s pattern similarity values were mean centered separately and were all included as predictors of boundary-related VTA activation in the same linear mixed effects model.

Comment #4: Fourth, because blink rate did not increase at event boundaries, it remains unclear whether the sharp VTA activation observed at those boundaries is directly tied to blink generation. Blinks might instead cluster before a boundary in anticipation, since the context

length was fixed at eight items and participants could easily predict when the next boundary would occur. I recommend analyzing the blink-rate time course at a finer temporal resolution to test for changes immediately before and after boundaries. Addressing these points would substantially strengthen the manuscript and clarify whether dopaminergic processes genuinely drive boundary-related distortions in event memory.

RESPONSE: This is an excellent point, given that event structure was consistent across all sequences. We believe there is both direct and indirect eye-tracking evidence that suggests that blinks don't cluster prior to the boundary but rather increase afterward. Furthermore, our pupil dilation data (primarily reported in Clewett et al., 2025; see Figures 4 and 5) also provide indirect support that participants could not fully anticipate the timing of the boundaries. We elaborate on these important datapoints below:

Event boundaries predicted a momentary increase in blinking after, not before, the boundary. When focusing on blink patterns at a finer temporal resolution (i.e., 1.5s on either side of the tones), we found:

“a main effect of Tone Type on local blink count ($\beta = .12$, $SE = .019$, $t(7461) = 6.07$, $p < .001$), such that boundary tones were immediately followed by more blinks than same-context tones” (Page 7).

Thus, boundary tones - which signaled task and perceptual switches - predicted increases in blinking in the 1.5s directly after each tone.

Importantly, we also performed a separate control analysis that was focused on the number of blinks leading up to the tones, enabling us to rule out the possibility that blinks were increasing in anticipation. Specifically, we examined whether Tone Type (boundary vs. same-context tone) or VTA activation predicted blinks in the 1.5s directly *before* each tone. We found that:

“pre-tone blink count was not significantly predicted by Tone Type ($\beta = .017$, $SE = .020$, $t(7410) = .83$, $p = .41$) or tone-related VTA activation ($\beta = .00012$, $SE = .00013$, $t(7224) = .95$, $p = .34$)” (Page 7).

We also underscore that while event length was fixed at eight items, creating a relatively predictable encoding structure, the ISIs between items were pseudo-randomly jittered to be 3s, 5s, or 7s. In this way, it was very difficult for participants to anticipate the precise timing of the boundaries.

Lack of anticipation-related pupil responses suggested that participants couldn't precisely predict event structure or boundary timings. Alongside our blink results, our pupil dilation data further support the idea that participants couldn't fully predict the moment an event boundary would occur (described in detail in a companion publication; Clewett et al., 2025). In a highly similar, behavior-only version of this study with fixed ISIs, it was shown that event boundaries triggered engagement of an early-peaking pupil component relating to temporal memory indices of event segmentation (Clewett et al., 2020). The early-peaking temporal profile suggests that this feature of pupil dilation signals the anticipation of an impending decision or motor response (Clewett, Gasser, & Davachi, 2020; Johansson et al., 2018; Steinhauer & Hakarem, 1992). However, when stimulus timings were unpredictable in the current version with variable ISIs, boundaries no longer significantly modulated loading onto the early-peaking pupil component (see Clewett et al., 2025, Figure 4). From the perspective that this aspect of pupil dilation normally reflects anticipatory processes, this null effect suggests that participants weren't better at predicting the timing of an impending boundary versus the repeated tones.

References

1. Antony, J. W., Hartshorne, T. H., Pomeroy, K., Gureckis, T. M., Hasson, U., McDougle, S. D., & Norman, K. A. (2021). Behavioral, physiological, and neural signatures of surprise during naturalistic sports viewing. *Neuron*, *109*(2), 377-390.
2. Baddeley, A. D. and Hitch, G. J. (1974). Working Memory. In Bower, G. (Ed.), *The Psychology of Learning and Motivation: Advances in Research and Theory*, Vol. 111. New York: Academic Press.
3. Buonomano, D. V., Buzsáki, G., Davachi, L., & Nobre, A. C. (2023). Time for memories. *Journal of Neuroscience*, *43*(45), 7565-7574.
4. Clewett, D., Huang, R., & Davachi, L. (2025). Locus coeruleus activation “resets” hippocampal event representations and separates adjacent memories. *Neuron*.
5. Clewett, D., Gasser, C., & Davachi, L. (2020). Pupil-linked arousal signals track the temporal organization of events in memory. *Nature Communications*, *11*(1), 4007.
6. Crottaz-Herbette, S., & Menon, V. (2006). Where and when the anterior cingulate cortex modulates attentional response: combined fMRI and ERP evidence. *Journal of Cognitive Neuroscience*, *18*(5), 766-780.
7. D'Esposito, M., & Postle, B. R. (2015). The cognitive neuroscience of working memory. *Annual Review of Psychology*, *66*(1), 115-142.
8. Demiral, Ş. B., Kure Liu, C., Benveniste, H., Tomasi, D., & Volkow, N. D. (2023). Activation of brain arousal networks coincident with eye blinks during resting state. *Cerebral Cortex*, *33*(11), 6792-6802.
9. DuBrow, S., & Davachi, L. (2013). The influence of context boundaries on memory for the sequential order of events. *Journal of Experimental Psychology: General*, *142*(4), 1277.
10. Ezzyat, Y., & Clements, A. (2024). Neural activity differentiates novel and learned event boundaries. *Journal of Neuroscience*, *44*(38).
11. Fung, B. J., Sutlief, E., & Shuler, M. G. H. (2021). Dopamine and the interdependency of time perception and reward. *Neuroscience & Biobehavioral Reviews*, *125*, 380-391.
12. Glasser, M. F., Sotiropoulos, S. N., Wilson, J. A., Coalson, T. S., Fischl, B., Andersson, J. L., ... & Wu-Minn HCP Consortium. (2013). The minimal preprocessing pipelines for the Human Connectome Project. *Neuroimage*, *80*, 105-124.
13. Johansson, R., Pärnamets, P., Bjernstedt, A., & Johansson, M. (2018). Pupil dilation tracks the dynamics of mnemonic interference resolution. *Scientific Reports*, *8*(1), 4826.
14. Mikhael, J. G., & Gershman, S. J. (2019). Adapting the flow of time with dopamine. *Journal of Neurophysiology*, *121*(5), 1748-1760.
15. Pu, Y., Kong, X. Z., Ranganath, C., & Melloni, L. (2022). Event boundaries shape temporal organization of memory by resetting temporal context. *Nature Communications*, *13*(1), 622.
16. Reimer, J., McGinley, M. J., Liu, Y., Rodenkirch, C., Wang, Q., McCormick, D. A., & Tolias, A. S. (2016). Pupil fluctuations track rapid changes in adrenergic and cholinergic activity in cortex. *Nature Communications*, *7*(1), 13289.
17. Seeley, W. W. (2019). The salience network: a neural system for perceiving and responding to homeostatic demands. *Journal of Neuroscience*, *39*(50), 9878-9882.
18. Sutton, S., Braren, M., Zubin, J., & John, E. R. (1965). Evoked-potential correlates of stimulus uncertainty. *Science*, *150*(3700), 1187-1188.
19. Zacks, J. M., Kurby, C. A., Eisenberg, M. L., & Haroutunian, N. (2011). Prediction error associated with the perceptual segmentation of naturalistic events. *Journal of Cognitive Neuroscience*, *23*(12), 4057-4066.

20. Zacks, J. M., & Sargent, J. Q. (2010). Event perception: A theory and its application to clinical neuroscience. In *Psychology of learning and motivation* (Vol. 53, pp. 253-299). Academic Press.
21. Zacks, J. M., Speer, N. K., Swallow, K. M., Braver, T. S., & Reynolds, J. R. (2007). Event perception: a mind-brain perspective. *Psychological Bulletin*, 133(2), 273.

Response to Reviewers

Responses to Reviewer #1

Comment: I believe that the authors did their best to address my comments. It looks like my feedback improved the scientific merit of the manuscript leading to new and interesting findings. My feedback also served to provide means for the authors to learn new methods and modify their approach. Thus I believe that the authors performed well scientifically and this manuscript deserves publication.

RESPONSE: Thank you!

Responses to Reviewer #2

Comment #1: The authors have thoroughly addressed my prior set of concerns and I'm ready to sign off on the paper. One very minor point: In the Supplementary Material, the authors say "Blinks in the earlier vs. later interval did not influence temporal memory predictions" – I would phrase this as "Blinks in the earlier vs. later interval did not differentially influence temporal memory predictions"

RESPONSE: We thank Reviewer #2 for pointing out the proper language to describe this null interaction effect. We have now corrected the text.

Comment #2: Another small note: In the rebuttal letter (in response to my comment #5) the authors say: "There were no significant differences between the relationships of LC and VTA activation at tone switches and DG pattern similarity for image pairs spanning those switches" ... but there were in fact differences, which are accurately described in the text, and also in the detailed response to this comment. So this is an issue with the text of the rebuttal but not an issue with the paper itself; there's no actionable item here.

RESPONSE: We thank Reviewer #2 for pointing this inconsistency between the rebuttal and text/detailed response. As you noted, a revision was not required.

Responses to Reviewer #3

Comment: The authors have carefully and thoroughly addressed all previous comments, providing additional analyses that strengthen the manuscript. The new results clarify that VTA activation and blink behavior relate specifically to the perceived time dilation between events, rather than to temporal order judgments or hippocampal representations. This provides novel evidence suggesting that dopaminergic processes play a key role in segmenting memories of distinct events.

RESPONSE: Thank you!